# GenDrugCLIP: A Generation-Augmented Framework for Contrastive Drug-Target Representation Learning

## Abstract

Virtual screening (VS) has become an indispensable component of early drug discovery, aiming to identify potential ligands for a given protein target. While CLIP-style methods (e.g., DrugCLIP) have emerged as a powerful solution by enabling efficient active compound retrieval through drug-target representation alignment, current models face two fundamental challenges: (1) the scarcity of true binding data for training limits coverage of diverse binding modes, and (2) the use of trivial negatives (molecules binding to other pockets) leads to a significant train-test domain gap. To address these challenges, we introduce GenDrugCLIP, a novel generation-augmented framework that repositions structure-based drug design (SBDD) models as controllable data engines. GenDrugCLIP implements a Generate-Filter-Score-Select pipeline to construct target-aware pseudo positives and hard negatives for triplet contrastive learning. Our approach not only expands the chemical space but also prevents the model from relying solely on trivial negatives. Extensive experiments on three benchmarks demonstrate that GenDrugCLIP achieves state-of-the-art performance, outperforming DrugCLIP by +7.66% in BEDROC and +7.45 in $EF_{0.5\%}$ on the DUD-E benchmark. Our work highlights the untapped potential of SBDD models as powerful data engines for representation learning, opening a new paradigm for data-efficient drug discovery.

## 1 Introduction

Virtual screening (VS) is a fundamental component of early drug discovery, aiming to identify potential candidate compounds from vast chemical libraries for a specific protein target (Lionta et al., 2014). Traditional VS methods (Friesner et al., 2004; Trott & Olson, 2010) rely on molecular docking, which uses physical force fields to predict the binding conformation and assess the binding affinity between a molecule and its target. However, these methods require extensive conformations sampling and scoring, making them difficult to scale to billion-compound libraries.

Deep learning has revolutionized VS by enabling end-to-end prediction of protein-ligand interactions. Among existing approaches, supervised methods, such as deep neural networks for binding pose prediction and affinity regression, have achieved strong performance by learning from 3D structural complexes or experimental affinity measurements (e.g., $IC_{50}$, $K_i$) (Öztürk et al., 2018; Yan et al., 2023). However, such supervision is scarce and costly to obtain, with public DTI datasets containing only tens of thousands of high-quality entries, leading to poor generalization (Yan et al., 2023). Recent work has shifted toward self-supervised paradigms. DrugCLIP (Gao et al., 2023b) reframes VS as a dense retrieval task. DrugCLIP uses a contrastive objective to align ligands with targets without relying on explicit binding affinity labels. This eliminates the need for costly annotations and enables efficient billion-scale screening through conformation-free inference via embedding similarity search, achieving strong performance with reduced computational cost.

Despite its promise, training CLIP-style models in drug discovery still faces two key challenges: (1) True ligands remain sparse. Although no carefully curated binding affinity labels are required, each target has only a handful of known ligands. This scarcity limits the model's exposure to diverse binding modes and generalizable interaction patterns (Gao et al., 2023a). (2) Negative sample construction is overly simplistic. In standard in-batch negative sampling strategy (Radford et al., 2021)

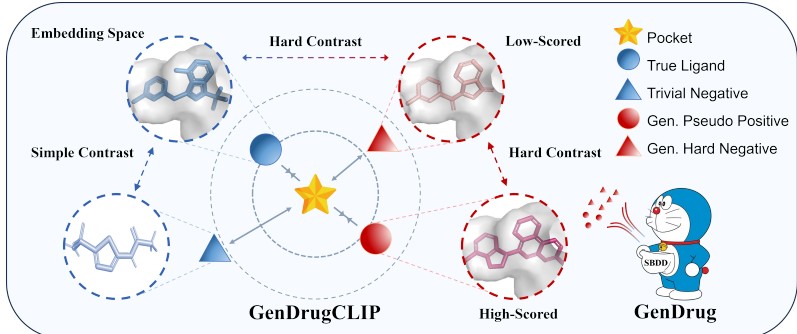

Figure 1: DrugCLIP uses standard CLIP-style contrastive learning to pull True Ligand closer and push Trivial Negatives apart from the pocket. GenDrugCLIP augments this pipeline with pocket-conditioned generated Pseudo Positives and Hard Negatives to expand chemical space coverage and construct hard contrast.

used in DrugCLIP training, for a given pocket, all ligands paired with other pockets in the same batch are treated as negative samples. However, these ligands are often chemically and structurally distinct to the true binders of the query pocket, making them "trivial negatives" that are easily distinguishable from positives. As illustrated in Figure 1, the contrastive learning process for a ligand-pocket pair includes one true ligand, and the negatives are molecules binding to other pockets with very different chemical sizes and structures. Prior research (Robinson et al., 2021; Huynh et al., 2022) have shown that using this kind of easy-to-distinguish negatives in contrastive learning can lead to "shortcut learning", where models rely on coarse-grained statistical differences rather than learning complex patterns, thus compromising performance.

In real-world VS, a model must not only identify novel and bindable molecules but also exclude those that are similar to known ligands yet fail to bind due to subtle conformational or chemical differences. The scarcity of these two critical sample types during training results in a significant train-test domain gap, which severely constrains the model's generalizability (Shen et al., 2025). A conventional approach to address this limitation is to augment the training data by perturbing known ligands. However, the perturbation process is **target-agnostic**. The resulting variants are often mere modifications of known ligands and may disrupt key binding interactions, rendering these augmented positives neither novel nor structurally valid. Concurrently, the undirected nature of such perturbations makes it difficult to precisely generate hard negatives that are similar to positive samples but exhibit hardly binding. Thus, there is an urgent need for a mechanism capable of constructing target-aware molecules to augment the training data.

We hypothesize that structure-based drug design (SBDD) models offer a promising solution. These models generate molecules conditioned on the protein binding pocket, demonstrating a unique potential to explore the **target-specific** chemical space (Tang et al., 2024). Although molecules generated by SBDD models may exhibit poor drug-likeness or low synthetic accessibility, which limits their direct use in drug design (Parrot et al., 2023; Schneuing et al., 2024), their generation process holds a crucial advantage: The generated molecules are intrinsically focused on the potential binding regions of the target, forming a curated set of novel, target-aware chemical compounds.

This insight leads us to reposition SBDD models as target-aware data engines for representation learning. We propose **GenDrug**, a Generate-Filter-Score-Select pipeline. GenDrug leverages SBDD model to produce pocket-conditioned molecules, followed by drug-likeness filtering and compatibility scoring to construct two crucial types of samples: (1) **Pseudo Positives:** High-scored molecules used to alleviate the issue of sparse true positives, (2) **Hard Negatives:** Molecules with low scores used to create challenging negative pairs.

Building on GenDrug, we introduce **GenDrugCLIP**, a contrastive learning framework that integrates generated samples into a triplet-based training set and performs hard negative mining. We employ a triplet contrastive loss (Patel et al., 2024) that draws strong binding molecules closer to its target while pushing away hardly binding decoys in shared embedding space. Figure 1 illustrates how GenDrugCLIP integrates pseudo positives and hard negatives into contrastive learning training process.

Our main contributions are summarized as follows:

1. We introduce GenDrugCLIP, a novel generation-augmented framework for contrastive drug-target representation learning. GenDrugCLIP leverages target-aware pseudo positives and hard negatives generated by GenDrug within a triplet-based contrastive learning objective to mitigate the sparsity of positives and overly simplistic negative construction.

2. Extensive experimental results show that GenDrugCLIP achieves state-of-the-art performance with significant improvements across three external test sets. Notably, it demonstrates +7.66% improvement in BEDROC and +7.45 in $EF_{0.5\%}$ on the DUD-E benchmark compared to DrugCLIP.

3. We demonstrate a new role for generative models in biomolecular representation learning: beyond generating candidates for synthesis, they can serve as data construction tools that characterize the binding-competent chemical space. This shifts the paradigm from viewing generation as an end goal to a means of scalable data augmentation, which is particularly valuable for understudied proteins in data-scarce regimes.

## 2 METHOD

Given a protein pocket $p$ and a set of candidate molecules $M = \{m_1, \ldots, m_a\}$, virtual screening aims to rank these molecules by their predicted probability of binding to $p$. We introduce GenDrugCLIP, a generation-augmented contrastive framework for drug-target representation learning. GenDrugCLIP optimizes a triplet contrastive loss on the training set expanded with GenDrug-generated molecules. The overall workflow is depicted in Figure 2 and detailed in the following subsections.

Section 2.1 briefly reviews the contrastive learning framework employed by DrugCLIP (Gao et al., 2023b) for completeness. We then introduce GenDrug, a target-aware generative data augmentation pipeline designed to produce pseudo positive and hard negative samples (Section 2.2). Finally, we present GenDrugCLIP, a novel contrastive learning framework that integrates these generated samples into the training pipeline to improve model robustness and generalization (Section 2.3).

### 2.1 CONTRASTIVE LEARNING-BASED DRUGCLIP

DrugCLIP formulates virtual screening as a cross-modal retrieval task, leveraging contrastive learning to align representations of protein pockets and ligand without relying on expensive binding affinity labels or pose optimization. Formally, let $\mathcal{P}$ and $\mathcal{M}$ denote the input spaces of pockets and molecules from training dataset PDBBind (Wang et al., 2005). DrugCLIP employs a dual-tower architecture with two modality-specific encoders $f_{\mathcal{P}} : \mathcal{P} \to \mathbb{R}^d$ and $f_{\mathcal{M}} : \mathcal{M} \to \mathbb{R}^d$ from Uni-Mol (Zhou et al., 2023) to map pocket $p \in \mathcal{P}$ and molecule $m \in \mathcal{M}$ into a $d$-dimensional shared embedding space:

$$z_p = f_{\mathcal{P}}(p), z_m = f_{\mathcal{M}}(m). \tag{1}$$

The similarity between them is measured using cosine similarity:

$$s(p, m) = \frac{z_p^T z_m}{||z_p|| \cdot ||z_m||}. \tag{2}$$

The model is trained using a bidirectional InfoNCE loss:

$$\mathcal{L}^{\mathcal{P}} = -\frac{1}{N} \sum_{i=1}^{N} \log \frac{\exp\left(s(p_i, m_i)/\tau\right)}{\sum_j \exp\left(s(p_i, m_j)/\tau\right)}, \tag{3}$$

$$\mathcal{L}^{\mathcal{M}} = -\frac{1}{N} \sum_{i=1}^{N} \log \frac{\exp\left(s(p_i, m_i)/\tau\right)}{\sum_j \exp\left(s(p_j, m_i)/\tau\right)}, \tag{4}$$

$$\mathcal{L} = \frac{1}{2}(\mathcal{L}^{\mathcal{P}} + \mathcal{L}^{\mathcal{M}}), \tag{5}$$

where $\tau$ is the trainable temperature parameter and $N$ is the batch size. For each pair $(p_i, m_i)$, all other molecules $m_j$ and pockets $p_j$ in the same batch are used as negative samples. To improve

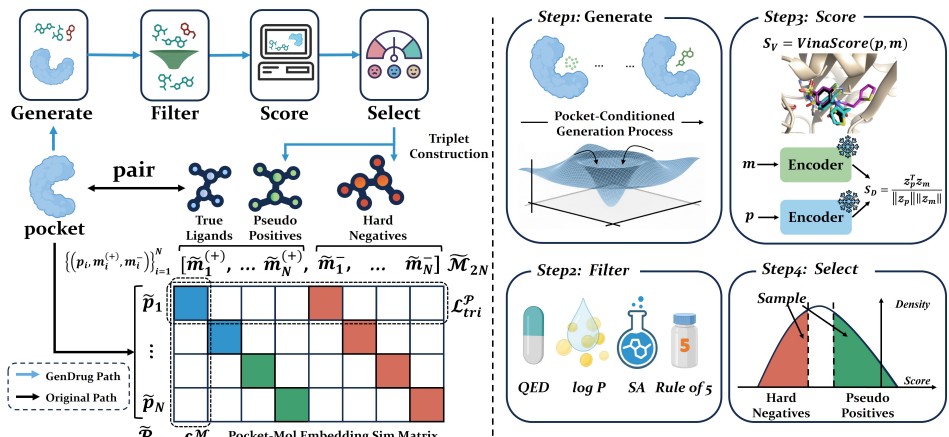

Figure 2: Illustration of GenDrugCLIP. Left: overall workflow and pocket-molecule embedding similarity matrix showing how GenDrug constructs (blue path) and supplies pseudo positives (green) and hard negatives (red). These generated samples are integrated into GenDrugCLIP training to optimize the model with a triplet loss. Right: four detailed steps of the Generate-Filter-Score-Select GenDrug workflow. It generates pocket-conditioned molecules, filters for drug-likeness, scores compatibility using physical affinity ($S_V$) or semantic similarity ($S_D$), and selects pseudo positives and hard negatives by hierarchical sampling.

robustness to structural variations in protein pockets, DrugCLIP introduces HomoAug, a biology-inspired augmentation strategy that derives pocket variants from homologous proteins. For each pocket $p_i \in \mathcal{P}$, three structurally similar variants $\{p_i^1, p_i^2, p_i^3\}$ are paired with the original ligand $m_i$, expanding the pocket space to $\mathcal{P}_{aug}$ and the number of pairs. In training, these four samples are treated equally. We denote the whole augmented training set as $\mathcal{D}_{ori}$. To ensure a fair comparison and maintain consistency with the original methods, we regard $\mathcal{D}_{ori}$ as the foundation for subsequent data supplementation and triplet construction in our framework.

## 2.2 GENDRUG: GENERATION-BASED DRUG DATA AUGMENTATION

To enrich the training data with high-quality, target-aware ligands, we propose **GenDrug**, a target-aware generative data augmentation pipeline. As illustrated in Figure 2 (right), GenDrug follows a four-stage Generate-Filter-Score-Select pipeline to construct samples. This pipeline enables the controlled generation of chemically valid and biologically relevant molecules and extends the data distribution beyond naturally observed binders. This pipeline is then integrated into GenDrugCLIP through blue path in Figure 2 (left). We detail each stage below.

**Step 1. Generating Molecule Candidates:** To generate structurally plausible molecules conditioned on target protein pockets, we adopt MolCRAFT (Qu et al., 2024), a structure-based drug design (SBDD) model based on Bayesian Flow Networks (BFNs), which generates molecules in a continuous latent space while preserving *SE*(3) equivariance. MolCRAFT is able to generate 3D ligand structures that adhere to the geometric constraints of the binding pocket, thereby minimizing conformational distortions and steric clashes. As a result, the generated molecules exhibit enhanced structural stability and biologically plausible binding modes. For each pocket in $\mathcal{D}_{ori}$, we generate 100 molecular candidates. On a single NVIDIA RTX 4090 GPU, this process takes approximately 120 seconds per pocket—over 30× faster than traditional diffusion-based models. This high through-put enables large-scale data augmentation.

**Step 2. Filtering for Drug-Likeness:** While SBDD models generate structurally feasible molecules for each pocket, some candidates may exhibit poor pharmaceutical properties. Including such molecules in training could introduce noise and degrade model performance. To ensure chemical validity and drug-likeness, we apply a filtering step based on four commonly used criteria: QED (Quantitative Estimate of Drug-likeness), SA (Synthetic Accessibility), LogP (Partition coefficient) and Lipinski's Rule of Five. In this work, we adopt moderate thresholds in these metrics. Only molecules satisfying all four criteria are retained for subsequent stages. This filtering step improves

the chemical validity and pharmaceutical relevance of the generated set, ensuring that the augmented data resides within realistic drug-like chemical space. We further discuss the impact of filter strategies in Appendix D.

**Step 3. Scoring Fitness:** After filtering, we independently evaluate the compatibility between each candidate molecule $m$ and its target pocket $p$ using two orthogonal scoring functions: a learned similarity score $S_D$ and a physics-based binding affinity score $S_V$, providing two perspectives on molecular. We describe the two scores below:

**The learned score** $S_D(p, m)$ is derived from the original DrugCLIP model. It measures the cosine similarity between the embeddings of the pocket and molecule: $S_D(p, m) = \frac{z_p^T z_m}{||z_p|| \cdot ||z_m||}$. This score reflects how closely the molecule aligns with the target pocket in the shared semantic space, capturing high-level structural and functional patterns associated with binding.

**The physics-based score** $S_V(p, m)$ is calculated using AutoDock Vina (Trott & Olson, 2010): $S_V(p, m) = VinaScore(p, m)$. $S_V$ estimates binding affinity through conformational search and empirical scoring functions based on molecular mechanics. Lower $S_V$ indicates stronger binding. $S_V$ evaluates the physical plausibility of interactions under geometric and energetic constraints.

**Step 4. Selecting Strategy:** Building on scoring outputs, we design a hierarchical sampling strategy to construct two types of high-quality training samples for contrastive learning: pseudo positives and hard negatives. Here, $S_D$ and $S_V$ are used independently to guide the sampling process. We next describe the selection criteria for each type of samples.

**Pseudo Positives:** We define the pseudo positive set $\mathcal{M}_i^+$ for each pocket $p_i$. For $S_D$, we select the top-$K$ molecules with the highest $S_D$. For $S_V$, we include all molecules with $S_V < \theta_{pos}$.

**Hard Negatives:** We define the hard negatives set $\mathcal{M}_i^-$ for each pocket $p_i$. We collect candidates with significantly worse scores (i.e., low rank in $S_D$ or $S_V > \theta_{neg}$) than those in $\mathcal{M}_i^+$ to form the hard negative set $\mathcal{M}_i^-$. Crucially, to avoid including ambiguous samples due to scoring inaccuracies, we introduce a gap zone which is an intermediate score margin that excludes molecules with moderate $S_D$ or $S_V$ values. This ensures that selected negatives are confidently hardly-binding and reduces noise from scoring model uncertainty.

## 2.3 GENDRUGCLIP

We propose **GenDrugCLIP**, a contrastive learning framework that leverages specially generated samples from GenDrug to improve representation learning. GenDrugCLIP's key ideas are: (1) supplementing pairs with pseudo positives to mitigate data sparsity, and (2) constructing challenging triplets with hard negatives to augment contrastive learning.

Let $D_{ori} = \{(p_i, m_i)\}_{i=1}^n$ denote the original training set, where each $(p_i, m_i)$ represents a known or homology-augmented protein–ligand pair. For each pocket $p_i$, we obtain two curated sets $\mathcal{M}_i^+$ and $\mathcal{M}_i^-$ from GenDrug. We sample from these sets to enrich training dataset through the following two steps.

**Pair Enrichment with Pseudo Positives:** For each pocket $p_i \in \mathcal{P}$, we sample $l$ molecules from $\mathcal{M}_i^+$ to form augmented positive pairs. This yields an extra positive pair set:

$$\mathcal{D}_{pos} = \{(p_i, m_{i,j}^+) \mid p_i \in \mathcal{P}, m_{i,j}^+ \in \mathcal{M}_i^+, j = 1 \dots l\}, \tag{6}$$

which is then merged with the original data to form the mixed dataset:

$$\mathcal{D}_{mix} = \mathcal{D}_{ori} \cup \mathcal{D}_{pos}. \tag{7}$$

We vary $l$ to systematically evaluate the effect of mixing ratio in ablation study. In subsequent sections, both experimentally verified binders and pseudo positives are treated equally as valid positive samples and denoted collectively as $m^{(+)}$.

**Triplet Construction with Hard Negatives:** We continue to construct triplet-based dataset based on $\mathcal{D}_{mix}$. For each pair $(p_i, m_i^{(+)})$ in $\mathcal{D}_{mix}$, we sample a corresponding hard negative molecule $m_i^-$

Table 1: Results on DUD-E in zero-shot setting. **Bold** indicates the best performance, underline denotes the second best.

| Method | AUROC(%) | BEDROC(%) | $EF_{0.5\%}$ | $EF_{1\%}$ | $EF_{5\%}$ |
|---|---|---|---|---|---|
| DeepDock* | 63.76 | 14.80 | 5.14 | 4.85 | 3.73 |
| pafnucy | 63.11 | 16.50 | 4.24 | 3.86 | 3.76 |
| dltlaVinaRF | 69.70 | 21.30 | 9.52 | 8.00 | 4.38 |
| NNscore2.0 | 68.30 | 12.20 | 4.16 | 4.02 | 3.12 |
| Kdeep* | 66.06 | 20.10 | 9.25 | 7.79 | 3.91 |
| PIGNET* | 67.36 | 19.90 | 7.90 | 6.72 | 3.88 |
| 3D-GNN* | 58.66 | 15.10 | 4.91 | 4.32 | 3.02 |
| RTMScore* | 65.84 | 29.50 | 10.60 | 9.32 | 5.31 |
| Glide SP | 76.70 | 40.70 | 19.39 | 16.18 | 7.23 |
| EquiScore | 77.57 | 43.23 | 20.94 | 17.67 | 7.82 |
| DrugCLIP | **78.68** | 37.76 | 28.18 | 23.48 | 8.73 |
| GenDrugCLIP$_{DC}$ | 77.80 | 43.60 | 34.99 | 28.07 | 9.38 |
| GenDrugCLIP$_{Vina}$ | 78.11 | **45.42** | **35.63** | **29.24** | **9.79** |

from $\mathcal{M}_i^-$ to construct a triplet $(p_i, m_i^{(+)}, m_i^-)$. This results in a triplet-based training set:

$$\mathcal{D}_{tri} = \{(p_i, m_i^{(+)}, m_i^-) \mid (p_i, m_i^{(+)}) \in \mathcal{D}_{mix}, m_i^- \in \mathcal{M}_i^-\}. \qquad (8)$$

**Training Objective:** To effectively leverage $\mathcal{D}_{tri}$, we extend the contrastive objective of Drug-CLIP. As illustrated in Figure 2 (left), each training batch samples $N$ triplets $\{(p_i, m_i^{(+)}, m_i^-)\}_{i=1}^N$. The protein and molecule encoders $f_\mathcal{P}$ and $f_\mathcal{M}$ map each entity to its embedding: $(\widetilde{p}_i, \widetilde{m}_i^{(+)}, \widetilde{m}_i^-)$. We then construct an extended molecule representation list by concatenating $2N$ molecule embeddings: $\widetilde{\mathcal{M}}_{2N} = [\widetilde{m}_1^{(+)}, \ldots, \widetilde{m}_N^{(+)}, \widetilde{m}_1^-, \ldots, \widetilde{m}_N^-] \in \mathbb{R}^{2N \times d}$ and a pocket representation list $\widetilde{\mathcal{P}}_N = [\widetilde{p}_1, \ldots, \widetilde{p}_N] \in \mathbb{R}^{N \times d}$. The extended similarity matrix is computed as: $S = \widetilde{\mathcal{P}}_N \cdot \widetilde{\mathcal{M}}_{2N}^T \in \mathbb{R}^{N \times 2N}$, where each row contains the cosine similarities between a pocket and $2N$ molecules in the embedding space.

We adopt a modified contrastive loss $\mathcal{L}_{tri}$. The denominator for the pocket-to-molecule direction loss $\mathcal{L}_{tri}^\mathcal{P}$ includes all positive and hard negative molecules and increases the difficulty of positive identification. We do not compute the symmetric molecule-to-pocket loss for hard negatives, as they have no true matching pockets:

$$\mathcal{L}_{tri}^\mathcal{P} = -\frac{1}{N} \sum_{i=1}^N \log \frac{\exp\left(s(p_i, m_i^{(+)})/\tau\right)}{\sum_j \exp\left(s(p_i, m_j^{(+)})/\tau\right) + \sum_j \exp\left(s(p_i, m_j^-)/\tau\right)}, \qquad (9)$$

$$\mathcal{L}^\mathcal{M} = -\frac{1}{N} \sum_{i=1}^N \log \frac{\exp\left(s(p_i, m_i^{(+)})/\tau\right)}{\sum_j \exp\left(s(p_j, m_i^{(+)})/\tau\right)}, \qquad (10)$$

$$\mathcal{L}_{tri} = \frac{1}{2}(\mathcal{L}_{tri}^\mathcal{P} + \mathcal{L}^\mathcal{M}). \qquad (11)$$

## 3 EXPERIMENTS

### 3.1 EXPERIMENTAL SETUP

**Training Dataset.** The original training data is primarily sourced from PDBBind (Wang et al., 2005). However, this dataset exhibits "hard overlap" or "soft overlap" that certain protein targets and their corresponding ground-truth ligands either appear directly in both train and test sets, or are highly similar across splits. Solely deduplicating by PDB ID risks data leakage, leading to an overestimation of model performance (Roy et al., 2015; Sieg et al., 2019; Su et al., 2020). To ensure rigorous evaluation, we follow EquiScore's (Cao et al., 2024) deduplication strategy: exclude all proteins in the training set that share the same UniProt ID with any target in DUD-E (Mysinger et al., 2012) or DEKOIS 2.0 (Bauer et al., 2013). This eliminates the possibility of hard overlap and mitigates soft overlap. Appendix C provides a detailed analysis of overlap across datasets and

Table 2: Results on DEKOIS 2.0.

| Method | AUROC(%) | BEDROC(%) | $EF_{0.5\%}$ | $EF_{1\%}$ | $EF_{5\%}$ |
|---|---|---|---|---|---|
| DeepDock* | 62.42 | 11.77 | 4.46 | 3.38 | 2.87 |
| pafnucy | 60.60 | 9.10 | 2.79 | 2.58 | 2.14 |
| dltlaVinaRF | 63.00 | 15.80 | 5.41 | 4.47 | 2.87 |
| NNscore2.0 | 64.60 | 11.60 | 3.10 | 3.25 | 2.58 |
| Kdeep* | 56.96 | 6.80 | 2.33 | 1.95 | 1.82 |
| PIGNET* | 65.20 | 18.92 | 6.20 | 5.34 | 3.78 |
| 3D-GNN* | 67.20 | 23.77 | 9.34 | 7.41 | 4.77 |
| RTMScore* | 67.60 | 35.25 | 12.92 | 12.12 | 4.99 |
| Glide SP | 74.20 | 37.80 | 13.63 | 12.10 | 6.26 |
| EquiScore | **82.09** | 46.03 | 19.10 | 16.83 | 8.27 |
| DrugCLIP | 78.54 | 41.22 | 15.08 | 14.06 | 7.87 |
| GenDrugCLIP$_{DC}$ | 77.02 | **49.12** | **19.40** | **17.59** | **8.59** |
| GenDrugCLIP$_{Vina}$ | 76.31 | 47.08 | 18.53 | 16.77 | 8.38 |

discusses its impact on generalization assessment. As a result, the number of pairs in our $\mathcal{D}_{ori}$ decreased by 20% compared to DrugCLIP. For each remaining pocket, we employ GenDrug (Section 2.2) and GenDrugCLIP (Section 2.3) to construct the final training dataset $\mathcal{D}_{tri}$.

**Baselines.** To evaluate the performance of our method, we assess the zero-shot performance of GenDrugCLIP$_{DC}$ (GenDrugCLIP with $S_D$) and GenDrugCLIP$_{Vina}$(GenDrugCLIP with $S_V$) on 3 virtual screen benchmarks: DUD-E (Mysinger et al., 2012), DEKOIS 2.0 (Bauer et al., 2013) and LIT-PCBA(Tran-Nguyen et al., 2020). We adopt eleven baseline methods: DeepDock (Méndez-Lucio et al., 2021), pafnucy (Stepniewska-Dziubinska et al., 2018), dltlaVinaRF (Stepniewska-Dziubinska et al., 2018), NNScore 2.0 (Durrant & McCammon, 2011), Kdeep (Jiménez et al., 2018), PIGNET (Moon et al., 2022), 3D-GNN (Moon et al., 2022), RTMScore (Shen et al., 2022), Glide SP (Friesner et al., 2004), EquiScore (Cao et al., 2024), DrugCLIP (Gao et al., 2023b). Among them, pafnucy, dltlaVinaRF, NNScore 2.0, Glide SP, EquiScore and DrugCLIP are evaluated under the same deduplication protocol as our model. For the remaining baselines, hard overlap exists between their training and test datasets. We report the results of test targets that do not appear in training set for these methods. These adjusted results are marked with asterisks (∗) next to their model names.

### 3.2 EVALUATION ON VIRTUAL SCREENING BENCHMARKS

**DUD-E Benchmark** (Mysinger et al., 2012): DUD-E (Directory of Useful Decoys, Enhanced) benchmark is a well-established dataset for evaluating virtual screening methods. It includes 102 protein targets and a total of 22,886 known active compounds, averaging 224 actives per target. For each active molecule, the dataset provides 50 carefully selected decoy molecules that share similar physicochemical properties but dissimilar in 2D topologies.

As shown in Table 1, the results clearly demonstrate that our method outperforms other paradigms. GenDrugCLIP$_{Vina}$ achieves state-of-the-art with average BEDROC of 45.42%, outperforming EquiScore and DrugCLIP by +2.19% and +7.66%. On all early enrichment metrics, GenDrug-CLIP achieves substantial improvements over both its base model DrugCLIP and EquiScore. Notably, DrugCLIP already demonstrates strong performance. GenDrugCLIP$_{Vina}$ further pushes the limit, surpassing EquiScore by +14.69 in $EF_{0.5\%}$. This highlights that our method significantly enhances the model's generalization performance on unseen targets. Our results also show that GenDrugCLIP$_{DC}$ achieves competitive results with over 10000× speedup in scoring process. This trade-off suggests that DrugCLIP score is a promising surrogate for fast screening, especially in large-scale training scenarios, while vina score remains valuable for final validation or small-scale, high-precision scoring.

**DEKOIS 2.0 Benchmark** (Bauer et al., 2013): DEKOIS 2.0 (Demanding Evaluation Kits for Objective In Silico Screening) is a widely used benchmark dataset for evaluating structure-based virtual screening methods. It comprises 81 protein targets across diverse families, with an average of approximately 128 active molecules per target.

Table 3: Results on LIT-PCBA.

| Method | AUROC(%) | BEDROC(%) | $EF_{0.5\%}$ | $EF_{1\%}$ | $EF_{5\%}$ |
|---|---|---|---|---|---|
| DrugCLIP | 54.93 | 3.78 | 3.43 | 2.96 | 1.69 |
| GenDrugCLIP$_{DC}$ | 56.71 | 5.03 | 5.64 | 3.98 | **2.19** |
| GenDrugCLIP$_{Vina}$ | **56.83** | **5.51** | **6.61** | **5.02** | 2.10 |

As shown in Table 2, GenDrugCLIP$_{DC}$ demonstrates consistent improvements over the base model DrugCLIP and achieves state-of-the-art performance. Notably, our method surpasses the current state-of-the-art EquiScore on BEDROC and all early enrichment metrics, which emphasize the prioritization of true actives in the very top ranks. This capability is critical for practical virtual screening. In metrics such as $EF_{0.5\%}$ and $EF_{1\%}$, GenDrugCLIP$_{Vina}$ performs on par with EquiScore, with negligible differences across targets. The results indicate that our method significantly enhances the model's sensitivity to high-value, hard-to-distinguish binders, making GenDrugCLIP a more effective solution for early-stage hit identification than its predecessor.

**LIT-PCBA Benchmark** (Tran-Nguyen et al., 2020): LIT-PCBA (Large-scale Interaction and Toxicity Prediction for Chemical Bioactivity Assays) comprises 15 protein targets 7844 active molecules and 407,381 unique inactive molecules. Crucially, LIT-PCBA is considered a more challenging benchmark than DUD-E or DEKOIS 2.0 due to its significantly higher proportion of non-binders, making the identification of true actives a more difficult task and providing a more rigorous test of a model's screening power.

In Table 3, we compare GenDrugCLIP with DrugCLIP under the same deduplication protocol to ensure a fair and stringent evaluation. GenDrugCLIP$_{Vina}$ demonstrates consistent performance gains across all evaluation metrics, with a notable improvement in the of +1.73%, +3.18 on BEDROC and $EF_{0.5\%}$. This trend is also observed in GenDrugCLIP$_{DC}$, indicating the robustness of the proposed approach.

We observed that the baseline DrugCLIP model exhibits varying performance across different datasets, specifically in terms of BEDROC scores: DEKOIS 2.0 > DUDE > LIT-PCBA. This indicates that DrugCLIP's intrinsic ability to discriminate between active and inactive molecules, and thus the reliability of its similarity scores, is highly context-dependent. In contrast, AutoDock Vina, as a general-purpose affinity estimation function, tends to offer a more robust and universally applicable measure of binding affinity across a wider range of chemical spaces and targets. Our results suggest that when the baseline DrugCLIP model demonstrates high performance on a dataset, leveraging its internal similarity metric for pseudo-label generation can further enhance performance. However, on datasets where DrugCLIP's performance is relatively low, using the more general and robust affinity estimator, Vina, is preferable. This understanding is crucial for selecting the appropriate affinity proxy function to optimize performance in specific drug discovery contexts.

### 3.3 ABLATION STUDY

**SBDD Model:** We conduct ablation studies on DUD-E benchmark by replacing the SBDD model with Targetdiff (Guan et al., 2023). We set up three distinct experiments. Targetdiff(pos-only): generate only positive samples using Targetdiff. Targetdiff(pos&neg): consistent with main model's setup but with a different SBDD model. MC (pos) & TD (neg): molecules generated by MolCRAFT are used as positive samples, while those generated by Targetdiff serve as negative samples.

By comparing the results of Targetdiff(pos&neg) and DrugCLIP, we can see that GenDrugCLIP pipeline still yield comparable improvements over baseline when utilizing the SBDD model TargetDiff. This result demonstrates the generalizability of our pipeline. Since the quality of molecules generated by TargetDiff is slightly weaker than that of MolCRAFT, the model's performance gain from positive samples is lower (Targetdiff(pos-only)). Intriguingly, the experiment MC(pos) & TD(neg) achieves results comparable to our main findings. This suggests that sampling negative examples from slightly weaker generators is also a viable strategy. By comparing the results of Targetdiff(pos&neg) and GenDrugCLIP$_{Vina}$, we can see that sample pseudo positives from stronger model improve overall performance, which implies more performance gain with stronger SBDD model in the future.

Table 4: Ablation Study on Various Aspects.”w/o” indicates the removal of the respective component compared to GenDrugCLIP$_{\text{Vina}}$.

| Ablation aspects | Method | AUROC(%) | BEDROC(%) | EF$_{0.5\%}$ | EF$_{1\%}$ | EF$_{5\%}$ |
|---|---|---|---|---|---|---|
| - | DrugCLIP | **78.68** | 37.76 | 28.18 | 23.48 | 8.73 |
| | GenDrugCLIP$_{\text{Vina}}$ | 78.11 | **45.42** | **35.63** | **29.24** | **9.79** |
| SBDD Model | Targetdiff(pos-only) | 75.69 | 38.68 | 30.67 | 24.38 | 8.57 |
| | Targetdiff(pos&neg) | 77.88 | 42.73 | 33.69 | 27.39 | 9.30 |
| | MC(pos) & TD(neg) | 77.73 | 45.27 | 36.23 | 28.93 | 9.73 |
| Filter | No Filtering | 78.35 | 43.88 | 33.98 | 27.93 | 9.60 |
| | Loose Filtering | 78.53 | 43.82 | 34.39 | 28.26 | 9.53 |
| Mixing Ratio | w/o mixing | 77.24 | 39.89 | 32.04 | 25.31 | 8.70 |
| | real:gen=4:1 | 78.48 | 44.69 | 34.76 | 28.64 | 9.72 |
| | real:gen=4:4 | 78.03 | 42.98 | 33.88 | 27.40 | 9.32 |
| Negative Strategy | w/o Neg | 76.12 | 38.74 | 30.66 | 24.01 | 8.61 |
| | DC+Neg | 77.24 | 39.89 | 32.04 | 25.31 | 8.70 |
| | with trivial Neg | 78.55 | 39.02 | 30.04 | 24.89 | 8.83 |

**Filter:** To investigate the impact of the 'Filter' stage, we conducted additional ablation experiments on DUD-E benchmark. These included scenarios where the filtering step was entirely removed ('No Filtering') and where a loose Vina score filtering criterion was applied (allowing approximately 90% of GT molecules to pass, termed 'Loose Filtering').

Our findings indicate that both 'No Filtering' and 'Loose Filtering' exhibited slightly weaker performance compared to the full pipeline in term of BEDROC and EF, which are important metrics for virtual screening. Nevertheless, they still significantly outperformed the baseline DrugCLIP model. This demonstrates that even with imperfect or less curated filtering criteria, the performance advantage of our method remains substantial, highlighting the robustness of our overall pipeline with respect to the specific filtering choices.

**Mixing Ratio:** We conduct an ablation study on the mixing ratio to investigate the impact of positive supplement strategy. The ratio effectively modulates how much the model relies on the enriched chemical space provided by generated molecules. In DrugCLIP, pockets are augmented with HomoAug, resulting in 4 entries for each pocket. We evaluate different levels of GenDrug-generated sample mixing ratio, specifically, w/o mixing(4:0), ratios of 4:1 and 4:4 (real:generated). Our results in Table 4 show that moderate injection (4:2, used in GenDrugCLIP$_{\text{Vina}}$) achieves the best performance, outperforming both 4:1 and 4:4. This indicates that GenDrug-generated samples provide valuable chemical diversity and structural novelty that complement the original data, enhancing the model's ability to generalize across unseen proteins. However, doubling the generated samples (4:4) leads to performance degradation, suggesting that excessive reliance on synthetic data may introduce distributional shift or noise that harms performance. The optimal balance at 4:2 reveals that generated positives are most effective when used to enrich rather than dominate the training, highlighting the importance of calibrated data augmentation.

**Hard Negative Strategy:** To evaluate the impact of our pocket-guided hard negative construction strategy, we conduct an ablation study on three settings of negative sampling schemes: "w/o Neg" (no negative mining), "DC+Neg" (DrugCLIP with negative mining), and "with trivial Neg" (replacement with random molecules from ZINC as negatives). The results in Table 4 demonstrate that the hard negative strategy consistently improves model performance over all metrics, whether applied to $D_{ori}$ directly ("DC+Neg" > DrugCLIP) or in conjunction with mixing strategy (GenDrugCLIP$_{\text{Vina}}$ > "w/o Neg"). The experiment "with trivial Neg" also demonstrates the importance of specially constructing the negatives. This highlights the significant benefit of tailoring target-aware negative samples to pockets in contrastive learning.

### 3.4 VISUALIZATION AND INTERPRETABILITY ANALYSIS

We visualize generated samples for the target 5H14 to illustrate the effectiveness of our Generate-Filter-Score-Select pipeline. As shown in Figure 3a, two generated pseudo positives (PP1, PP2) and two hard negatives (HN1, HN2) share a similar substructure encapsulated within the binding pocket, while differing in peripheral regions. Two PPs (PP1: $S_V = -10.79$, PP2: $S_V = -12.63$)

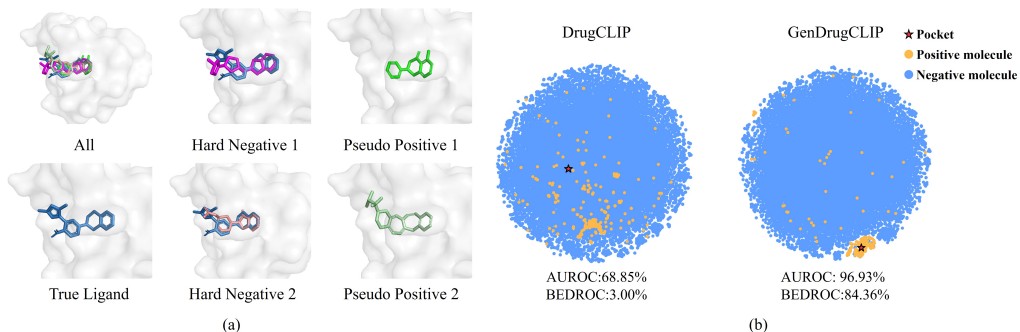

Figure 3: (a) Visualization of GenDrug-generated samples for the pocket of 5H14. (b) The t-SNE visualization of feature distributions for the target DPP4 from DUD-E using DrugCLIP and Gen-DrugCLIP.

exhibit strong docking scores, indicating their potential to enrich the positive chemical space. In contrast, two HNs (HN1: $S_V = -4.36$, HN2: $S_V = -3.72$) exhibit relatively low docking scores, making them structurally plausible and thus challenging for the contrastive model to distinguish. This demonstrates that GenDrug successfully generates and selects structurally diverse, target-aware compounds which are ideal for improving GenDrugCLIP's triplet contrastive learning.

We also perform a t-SNE visualization of the learned protein-ligand embedding space on the DPP4 target from DUD-E benchmark. As shown in Figure 3b, active ligands and decoys exhibit a clear separation in the embedding space, with active molecules clustering more tightly around the representation of the corresponding protein pocket. In contrast, embeddings generated by DrugCLIP show only moderate clustering of active ligands, but with a noticeable deviation from the target protein's embedding vector. These results suggest that our method achieves more accurate feature alignment and more discriminative representation learning in modeling protein-ligand interaction.

To evaluate the effectiveness of our proposed GenDrugCLIP, we embedded all molecules in the test dataset using both DrugCLIP and GenDrugCLIP. Subsequently, we computed the average similarity scores for Pocket-Active, Pocket-Decoy, and Active-Decoy pairs across all tested targets. For DrugCLIP, $Sim(P, A) = 0.3575$, $Sim(P, D) = 0.2021$, $Sim(A, D) = 0.1757$. for GenDrugCLIP, $Sim(P, A) = 0.3164$, $Sim(P, D) = 0.0997$, $Sim(A, D) = 0.0396$. The data demonstrates that GenDrugCLIP's architectural strategy optimizes the embedding space. While still achieving a robust similarity for Pocket-Active pairs, GenDrugCLIP markedly enhances the discrimination of subtly different decoy molecules. This is achieved by effectively learning to push decoy molecules further away from the binding pocket representations, thereby reducing the likelihood of false positives. The Active-Decoy similarity dramatically decreases from 0.1757 (DrugCLIP) to 0.0396 (GenDrug-CLIP). This significant reductio enhances separability between active compounds and their inactive decoys. This outcome demonstrates that GenDrugCLIP's strategy empowers the model to discern more subtle and fundamental distinctions between true active binders and structurally similar, yet inactive, decoy molecules.

## 4 CONCLUSION

We introduce GenDrugCLIP, a novel generation-augmented framework for contrastive drug-target representation learning. GenDrugCLIP leverages target-aware pseudo positives and hard negatives generated by GenDrug within a triplet-based contrastive learning objective to mitigate the sparsity of positives and overly simplistic negative construction. Importantly, our work repositions generative models not merely as tools for molecular design, but as controllable data engines that actively enrich the training samples for representation learning. This paradigm shift opens new avenues for efficient learning in data-scarce settings. We envision that generated samples can be used to provide plausible positive pairs and informative hard negatives for understudied targets, thereby facilitating candidate screening under few-shot or zero-shot settings. We believe the integration of generative modeling and self-supervised representation learning will inspire future work at the intersection of AI drug discovery and foundation model pretraining.

ETHICS STATEMENT

This work complies with the ICLR Code of Ethics. We have considered potential societal impacts, fairness, and transparency. No human subjects or sensitive data were involved. All datasets are publicly available and used in accordance with their licenses. We aim to promote responsible research and acknowledge the importance of minimizing harm and avoiding misuse.

REPRODUCIBILITY STATEMENT

We provide detailed descriptions of our methods, model architectures, and hyperparameters in the main body and appendix. All experiments are conducted using publicly available codebases, and we will release our code and trained models to ensure full reproducibility.

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

## A  LLM USAGE STATEMENT

We used a large language model (LLM) during the preparation of this manuscript to assist with language editing and proofreading. The model was used solely for improving the clarity and readability of the text. It did not contribute to the research design, idea development, data analysis, or technical content of the paper. All final content and scientific decisions were made and approved by the human authors.

## B  RELATED WORK

**Virtual Screening:** Traditional virtual screening relies on molecular docking tools like Glide (Friesner et al., 2004) and AutoDock (Trott & Olson, 2010), which use physics-based scoring functions to predict binding poses and affinities. Deep learning has significantly advanced virtual screening by enabling end-to-end modeling of protein-ligand interactions. Most of these methods train a deep model in a supervised way to predict the binding affinity between a molecule and a pocket (Öztürk et al., 2018; Shen et al., 2022; Moon et al., 2022) or calculating binding scores from 3D protein-ligand complex structures (Zhang et al., 2023; Cai et al., 2024). Recently, DrugCLIP (Gao et al., 2023b) introduces a contrastive retrieval paradigm, aligning ligand and pocket embeddings for billion-scale similarity search, shifting the paradigm from scoring to embedding-based screening.

**Structure-Based Drug Design:** Generative models for SBDD aim to design molecules conditioned on target pockets. Autoregressive approaches generate atoms sequentially with $SE(3)$ equivariance (Masuda et al., 2020; Luo et al., 2021), but suffer from high computational cost. Fragment-based methods improve efficiency (Powers et al., 2022; Zhang & Liu, 2023), yet require post-processing to correct assembly errors. Non-autoregressive models, particularly diffusion models (Guan et al., 2023; Schneuing et al., 2024) and Bayesian Flow Networks (BFNs) (Qu et al., 2024) improve sampling efficiency and quality. Some further incorporate affinity or property guidance to steer synthesis toward viable candidates (Huang et al., 2024).

Table 5: Test results of DrugCLIP under different remove setting

| Test Set | Method | AUROC(%) | BEDROC(%) | $EF_{0.5\%}$ | $EF_{1\%}$ | $EF_{5\%}$ |
|---|---|---|---|---|---|---|
| Lit-PCBA | DrugCLIP$_{dedup}$ | 54.93 | 3.78 | 3.43 | 2.96 | 1.69 |
| | Remove Non-leak | 59.02 | 6.79 | 8.53 | 5.85 | 2.46 |
| | Remove Globally | 58.44 | 5.70 | 6.59 | 4.76 | 2.07 |
| | DrugCLIP | 57.17 | 6.23 | 8.56 | 5.51 | 2.27 |
| DUD-E | DrugCLIP$_{dedup}$ | 74.84 | 32.68 | 25.19 | 20.4 | 7.61 |
| | Remove Non-leak | 81.28 | 45.27 | 34.01 | 28.48 | 10.25 |
| | Remove Globally | 78.53 | 44.74 | 35.15 | 28.54 | 9.68 |
| | DrugCLIP | 80.93 | 50.52 | 38.07 | 31.89 | 10.66 |

## C  OVERLAP BETWEEN TRAINING AND TESTING DATA

Similar to Equiscore, we conducted an analysis on overlap between training and testing data in Drug-CLIP. While the original training and test sets (DrugCLIP is evaluated on DUD-E and LIT-PCBA) are constructed to avoid exact PDB ID overlap which ensures independence at the level of experimentally resolved structures, further analysis reveals a deeper layer of contamination: approximately 20% of the training samples share the same UniProt ID as proteins in the test set. This indicates that the training data includes the same (or highly similar sequence variants/homologs) proteins to target protein present in the test set. Among these overlapping entries, we find that 5% samples contain ligands identical to active compounds in the test set (based on standardized SMILES), and 28% samples contain highly similar molecules (Tanimoto similarity > 0.8). This overlap in both target identity and ligand structure constitutes a significant information leakage, which enables models to recognize test instances through memorization rather than generalization. Such data overlap is inconsistent with the assumptions underlying the zero-shot evaluation protocol, which requires that test targets be entirely novel with respect to the training set. To align the benchmark with this principle, we follow EquiScore to apply a systematic decontamination procedure to the original training

set by excluding all protein-ligand pairs associated with any UniProt ID present in the test set. This ensures strict separation at the target level.

To isolate the impact of this data leakage, we conduct two controlled ablation studies:

- Randomly removing the same number of non-leaking samples (i.e., those with no UniProt overlap), and

- Randomly removing an equivalent number of samples globally.

In both cases, we retrain DrugCLIP on the reduced datasets while matching the final training size to that of the deduplication set. Results in Table 5 show that performance decreases only marginally in both control settings, indicating that the mere reduction in training data volume has limited effect. In contrast, after removing all samples with shared UniProt IDs, DrugCLIP$_{\text{dedup}}$ model performance drops significantly by 17.84% and 2.45% in BEDROC on DUD-E and LIT-PCBA, respectively.

This stark contrast demonstrates that overlapping protein–ligand pairs, particularly those involving shared targets and known active compounds, disproportionately contribute to the model's performance. Their presence introduces a favorable shortcut, enabling models to rely on memorization rather than learning transferable binding patterns.

## D    DISCUSSION ON THE IMPACT OF FILTERING

In this work, we apply moderately permissive filtering criteria to eliminate chemically unstable or synthetically intractable molecules and to reduce the computational burden of downstream Vina scoring. We apply the following thresholds: $QED > 0.4, SA > 0.5, 5 > LogP > 0.4$ and no more than one violation of Lipinski's Rule of Five. While these filters improve the overall quality of the generated set and improve the performance slightly (Table 6 "No Filtering" vs. GenDrugCLIP $_{\text{DC}}$), we emphasize that they are not strictly necessary for early-stage virtual screening. In real-world drug discovery pipelines, oral bio-availability and other related properties are often relaxed during initial hit identification, where the primary goal is to identify hits with binding potential rather than fully optimized leads. Importantly, our framework reveals that such filtering steps can be repurposed as design levers to encode inductive biases into the training data. By customizing the filtering rules and pairing them with strategic negative sampling, one can explicitly shape the model's sensitivity to specific molecular properties. For instance, enriching high-SA compounds in the positive set while pairing them with low-SA negatives would encourage the model to learn a decision boundary sensitive to synthetic complexity. Similarly, one could promote or suppress other traits—such as solubility, metabolic stability—by aligning filter design with data construction. This suggests a broader opportunity: beyond mere data cleaning, filtering criteria can serve as a mechanism for preference injection in self-supervised or contrastive learning frameworks.

## E    ADDITIONAL ABLATION STUDY

We further conduct ablation studies under the $S_D$ training setting (Table 6), evaluating the same data composition strategies: varying the mixing ratio of real to generated positives and applying the pocket-guided hard negative sampling. The results show trends consistent with those observed in $S_V$, where moderate data enrichment (e.g., 4:2 ratio) and hard negative construction lead to the best performance. This confirms the robustness and generalizability of our two core strategies across different training paradigms. Additionally, we evaluate a variant where GenDrug generates molecules without applying any post-generation filtering, and all samples are directly scored and selected based on $S_D$ . The resulting performance is lower compared to the filtered counterpart, indicating that quality filtering, although permissive, plays a non-trivial role in maintaining data integrity and enhancing model learning.

## F    IMPLEMENTATION DETAILS

We adopt MolCRAFT as the structure-based drug design (SBDD) model, generating 100 molecules per target binding pocket. For computing $S_D$, we use the trained DrugCLIP model. Pseudo positives

Table 6: Ablation Study on Various Aspects on $S_V$.

| Ablation aspects | Method | AUROC(%) | BEDROC(%) | $EF_{0.5\%}$ | $EF_{1\%}$ | $EF_{5\%}$ |
|---|---|---|---|---|---|---|
| - | DrugCLIP | **78.68** | 37.76 | 28.18 | 23.48 | 8.73 |
| | GenDrugCLIP$_{DC}$ | 77.80 | **43.60** | **34.99** | **28.07** | **9.38** |
| Mixing Ratio | real:gen=4:1 | 78.48 | 44.69 | 34.76 | 28.64 | 9.72 |
| | real:gen=4:4 | 78.03 | 42.98 | 33.88 | 27.40 | 9.32 |
| Negative Strategy | w/o Neg | 76.12 | 38.74 | 30.66 | 24.01 | 8.61 |
| | DC+Neg | 77.24 | 39.89 | 32.04 | 25.31 | 8.70 |
| Filtering Strategy | No Filtering | 77.37 | 41.06 | 33.13 | 26.11 | 8.91 |

are select by Top-10 $S_D$ and hard negatives by Top 20-40 $S_D$. The $S_V$ score is computed directly using AutoDock Vina (v1.2.2) with default parameters. Pseudo positives and hard negatives are select by $S_V < \theta_{pos}, S_V > \theta_{neg}$, respectively. In this work, $\theta_{pos} = -10$ and $\theta_{neg} = -8$. We use four-step gradient accumulation to train the model on a single GPU, enabling optimization equivalent to multi-GPU training using four devices. Inference and scoring are performed on NVIDIA GeForce RTX 4090 GPUs and Intel Core i9-13900K CPUs. During training, we sample 48 protein–molecule triplets (or pairs) per batch. All models are trained on NVIDIA RTX A6000 GPUs.

