# OpenReview forum: "GenDrugCLIP: A Generation-Augmented Framework for ContrastiveDrug-Target Representation Learning"
_ICLR.cc/2026/Conference — Submitted to ICLR 2026_

### Official Review · Reviewer_8bmZ · 2025-10-27

**Soundness:** 2
**Presentation:** 2
**Contribution:** 2
**Rating:** 4
**Confidence:** 5

**Summary:**

This paper introduces GenDrugCLIP, an extension of DrugCLIP, which incorporates a structure-based drug design (SBDD) model, MolCRAFT, to generate molecule samples conditioned on the target. These generated samples are then filtered and used as pseudo positives and negatives for data augmentation, aiming to enhance the training of DrugCLIP. While the method is conceptually reasonable, it achieves only marginal improvements on virtual screening benchmarks.

**Strengths:**

1. The idea of using a generative model to create pseudo hard negative samples for virtual screening is both interesting and conceptually sound.
2. The filtering strategy is thoughtfully designed and appears to be well-justified.
3. The proposed method leads to some kind of improvements over baselines.
4. The improvement over the “with trivial negatives” setting in the ablation study is encouraging, suggesting that generating negatives via a generative model is more effective than using random samples, although more details should be included.

**Weaknesses:**

1. The major concern of this paper is the relatively limited overall performance improvement. Both the enrichment factor and BEDROC show less than 10% gain, which may not be sufficient to convincingly demonstrate the effectiveness of the proposed method.
2. The overall techniqual depth and contribution of this paper is limited. It is based on an existing model for virtual screening, and use an existing  SBDD model for data augmentation.
3. The paper currently explores only one SBDD model: MolCRAFT. It is better to try more generative models and compare the performance.
4. The ablation study should be more detailed and thorough. For example, It is unclear whether the “trivial negatives” were also passed through the same filtering process, which may be a key factor behind the method’s success. If the random negatives were not filtered similarly, the comparison could be misleading. Clarifying this would strengthen the credibility of the analysis.
5. While the inclusion of visualizations is appreciated, Figure 3b provides limited insight. It only shows one case where GenDrugCLIP outperforms the original DrugCLIP on a specific target, without offering any analysis of the underlying causes. Although cherry-picking cases can illustrate potential improvements, a deeper explanation of why the improvement occurs is needed to enhance the informativeness of the result.

**Questions:**

see weakness

---

> ### Author Response · Authors · 2025-11-20
> **Response to Weaknesses 1&2**
>
> Thank you for your careful consideration and perceptive comments. We greatly appreciate your valuable questions and provide our responses below.
>
> >**W1: Limited overall performance improvement.**
>
> **A1:** We appreciate your conmment regarding the performance improvements. While percentage might seem modest, we argue that these improvements are both statistically significant and practically valuable within the context of drug discovery. BEDROC and EF are critical metrics specifically designed to evaluate model's ability to prioritize active compounds at the very top of a ranked list. In drug discovery, the goal is to identify a small number of true positives from a vast pool of candidates as early as possible. Therefore, even a small improvement in these metrics translates to a more efficient and cost-effective screening. Our model demonstrates significant absolute and relative improvements in both BEDROC and EF, which are crucial indicators of early enrichment capabilities. These improvements are achieved under a strict deduplication protocol, ensuring that the gains stem from the model's enhanced ability to generalize rather than overfitting to leaked data.
>
> The performance enhancements have direct and substantial implications for real-world drug discovery. An improvement in early enrichment means that for the same experimental effort (e.g., testing the top 0.5% of compounds), our model is likely to identify significantly more active compounds. For instance, our method improves the $EF_{0.5\\%} $ on the DUDE from a baseline of 28.18% to 35.63%(+7.45%). In an ideal scenario, this means discovering 26% more active compounds for the same wet-lab investment. Conversely, to get the same number of active compounds, our model would require testing 20% fewer molecules. Crucially, by integrating comprehensive drug-likeness and other pharmaceutical property knowledge into our sample selection process, the compounds identified by our model are\also more likely to possess desirable drug-like characteristics, making them more promising candidates for further development.
>
> >**W2:The overall techniqual depth and contribution of this paper is limited. It is based on an existing model for virtual screening, and use an existing SBDD model for data augmentation.**
>
> **A2:** We appreciate your attention to novelty. We wish to clarify that our work embodies a paradigm shift and holds implications far beyond traditional data augmentation. It directly addresses critical needs within the domain of drug discovery and possesses immense potential. We would like to show the significance of our method through the following two points:
>
> 1.**"Learning from synthetic data"** has emerged as a rapidly developing and highly influential research theme. Numerous studies are dedicated to enhancing model in data-scarce domains through synthetic data generation and augmentation strategies in many fields such as robotics[1] and medical imaging[2]. GenDrugCLIP represents a concrete and innovative instantiation of this trend within drug discovery. By integrating structural biology information into the generation process, it provides high-quality structured knowledge for protein-ligand representation learning. This distinguishes our approach from generic data augmentation; it constitutes a novel paradigm of domain-knowledge-driven synthetic data for representation learning.
>
> 2.**GenDrugCLIP is a complete, closed-loop representation learning framework.** The modular design allows for flexible substitution of its components with more advanced models, adapting to diverse practical scenarios. GenDrugCLIP's methodology is orthogonal to existing CLIP-style VS models (e.g., LigUnity[3]) and optimized methods (e.g., DrugHush[4]), suggesting significant potential for synergistic integration. More profoundly, GenDrugCLIP's potential lies in offering solutions for few-shot and zero-shot learning challenges in drug screening. By leveraging the principle of "learning from synthetic data," our approach can generate "virtual" ligands highly matched to novel targets. This effectively augments the available training data, thereby enhancing the model's learning capabilities in these challenging scenarios. Such an advancement holds immense practical significance for accelerating new drug development.
>
> **References**
>
> [1] Chang, Mincheol, et al. "Just Add $100 More: Augmenting Pseudo-LiDAR Point Cloud for Resolving Class-imbalance Problem." NeurIPS 2024.
>
> [2] Li, Jolina, et al. "Infusing synthetic data with real-world patterns for zero-shot material state segmentation." NeurIPS 2024.
>
> [3] Feng, Bin, et al. "Hierarchical affinity landscape navigation through learning a shared pocket-ligand space." Patterns (2025).
>
> [4] Han, Jin, Yun Hong, and Wu-Jun Li. "DrugHash: Hashing Based Contrastive Learning for Virtual Screening." AAAI 2025.

---

> ### Author Response · Authors · 2025-11-20
> **Response to Weaknesses 3&4**
>
> >**W3:The paper currently explores only one SBDD model: MolCRAFT. It is better to try more generative models and compare the performance.**
>
> **A3:** Thank you for this valuable point. MolCRAFT offers significant advantages, being **over 30 times faster** than traditional diffusion-based models and **the first to achieve reference-level Vina Score** generation quality with comparable molecular size. These characteristics make MolCRAFT suitable for large-scale, high-quality generation. Furthermore, GenDrugCLIP is designed as a general and modular framework. MolCRAFT serves as an excellent **proof-of-concept** for demonstrating the power of repositioning SBDD models as data engines.
>
> To further address your concern and comprehensively validate the generality of our pipeline, **we conducted additional ablation studies on DUD-E benchmark by replacing the SBDD model with Targetdiff.** We set up three distinct experiments: 1. Targetdiff (pos-only): generate only positive samples using  Targetdiff. 2. Targetdiff (pos&neg): consistent with our main model's setup, but with a different SBDD model. 3. MC (pos) & TD (neg): molecules generated by MolCRAFT are used as positive samples, while those generated by Targetdiff serve as negative samples. $GenDrugCLIP_{Vina}$ refers to our main model. The experimental results are presented in the table below:
>
> | Ablation | Method | AUROC(%) | EDROC(%) | EF0.5% | EF1% | EF 5% |
> | :---: | :--- | :---: | :---: | :---: | :---: | :---: |
> | – | DrugCLIP | **78.68** | 37.76 | 28.18 | 23.48 | 8.73 |
> | – | $GenDrugCLIP_{Vina}$ | 78.11 | **45.42** | 35.63 | **29.24** | **9.79** |
> | SBDD | Targetdiff(pos-only) | 77.59 | 38.74 | 30.55 | 24.36 | 8.69 |
> |  | Targetdiff(pos&neg) | 77.88 | 42.73 | 33.69 | 27.39 | 9.30 |
> |  | MC(pos) & TD(neg) | 77.73 | 45.27 | **36.23** | 28.93 | 9.73 |
>
> By comparing the results of Targetdiff(pos&neg) and DrugCLIP, we can see that GenDrugCLIP pipeline still yield comparable improvements over baseline when utilizing the SBDD model TargetDiff. This result demonstrates the generalizability of our pipeline. Since the quality of molecules generated by TargetDiff is slightly weaker than that of MolCRAFT, the model's performance gain from positive samples is lower (Targetdiff(pos-only)). Intriguingly, the experiment MC(pos) & TD(neg) achieves results comparable to our main findings. This suggests that sampling negative examples from slightly weaker generators is also a viable strategy. By comparing the results of Targetdiff(pos&neg) and $GenDrugCLIP_{Vina}$, we can see that sample pseudo positives from stronger model improve overall performance, which implies more performance gain with stronger SBDD model in the future.
>
> >**W4: The ablation study should be more detailed and thorough. For example, It is unclear whether the “trivial negatives” were also passed through the same filtering process, which may be a key factor behind the method’s success. If the random negatives were not filtered similarly, the comparison could be misleading. Clarifying this would strengthen the credibility of the analysis.**
>
> **A4:** We agree that ablation study should be more detailed in core components. The relevant experiments should prove that using tailored target-aware negative samples to pockets in this article is the key to the success of the method, compared to using random molecules or other methods. The 'trivial negatives' attempt to prove this, but without stripping away the influence of multiple filters in this process. To address this, we conduct an additional ablation study, which we refer to as Setting 1.
>
> In this Setting 1, we randomly sample molecules from ZINC (match the ligand atom numbers to reference ligands) to ensure a fair comparison in terms of molecular size. These randomly sampled molecules are subjected to the same property filters and DrugCLIP scoring as applied in our main framework. The key distinction between Setting 1 and GenDrugCLIP_DC lies solely in the source of data. This allows us to directly assess the value added by MolCRAFT's generative capabilities prior to filtering. The results for this new Setting 1 are presented below:
>
> | Method | AUROC(%) | EDROC(%) | EF0.5% | EF1% | EF 5% |
> | :--- | :---: | :---: | :---: | :---: | :---: |
> | DrugCLIP | 78.68 | 37.76 | 28.18 | 23.48 | 8.73 |
> | $GenDrugCLIP_{DC}$ | 77.80 | 43.60 | 34.99 | 28.07 | 9.38 |
> | Setting1 | 75.69 | 38.68 | 30.67 | 24.39 | 8.57 |
>
> The results indicate that the model based on randomly selected molecule Setting 1 has almost no improvement compared to the baseline model, and randomly selected molecules are difficult to provide effective learning information to the model without pocket information.

---

> ### Author Response · Authors · 2025-11-20
> **Response to Weakness 5**
>
> >**W5: While the inclusion of visualizations is appreciated, Figure 3b provides limited insight. It only shows one case where GenDrugCLIP outperforms the original DrugCLIP on a specific target, without offering any analysis of the underlying causes. Although cherry-picking cases can illustrate potential improvements, a deeper explanation of why the improvement occurs is needed to enhance the informativeness of the result.**
>
> **A5:** We acknowledge that cherry-picking cases without deeper analysis reduces informativeness. To address this and provide a more comprehensive insight into our method's mechanism, we conducted an additional analysis to demonstrate how GenDrugCLIP optimizes its representation space. Specifically, we used both DrugCLIP and GenDrugCLIP to embed all molecules in the test dataset. We then calculated the average similarity between three critical pairs of entities: Pocket-Active molecules, Pocket-Decoy molecules, and Active-Decoy molecules.
>
> | | Pocket-Active | Pocket-Decoy | Active-Decoy |
> | :--- | :--- | :--- | :--- |
> | DrugCLIP | 0.3575 | 0.2021 | 0.1757 |
> | GenDrugCLIP | 0.3164 | 0.0997 | 0.0396 |
>
> The results indicate that GenDrugCLIP's construction strategy optimizes the embedding space: while ensuring high similarity for pocket-actives, GenDrugCLIP enhances the discrimination of decoy molecules with subtle differences, effectively learning to push decoy molecules away from the binding pocket. The similarity between active molecules and decoy molecules (Active-Decoy) significantly decreased, improving the separability between active and decoy molecules. This demonstrates that GenDrugCLIP's strategy enables the model to recognize more subtle and essential differences between true active binders and subtly different inactive decoy molecules.

---

### Official Review · Reviewer_we3Y · 2025-10-28

**Soundness:** 2
**Presentation:** 3
**Contribution:** 2
**Rating:** 4
**Confidence:** 4

**Summary:**

This paper proposes a generation augmented contrastive learning framework tGenDrugCLIP, to address two key limitations of CLIP-style virtual screening: (1) sparse true binding data and (2) reliance on trivial negatives. By repurposing SBDD models as controllable data engines, it introduces a Generate-Filter-Score-Select pipeline to create target-aware pseudo positives and hard negatives for triplet contrastive learning.

**Strengths:**

1. This study successfully applies Structure-Based Drug Design (SBDD) to new practical scenarios(Virtual Screening), yielding positive effects.

2. The paper is well structured and easy to follow.

3. The paper evaluates on three benchmarks: DUD-E, DEKOIS 2.0, and LIT-PCBA. On DUD-E, GenDrugCLIPVina achieves 45.42 % BEDROC (+7.66 % over DrugCLIP) and EF0.5 % of 35.63 (+7.45 over DrugCLIP). It also sets new state-of-the-art on DEKOIS 2.0 (BEDROC 49.12 % ) and improves over DrugCLIP on the challenging LIT-PCBA dataset (BEDROC 5.51 % vs. 3.78 %).

**Weaknesses:**

Concerns:

1. AutoDock Vina’s known biases (e.g., molecular-weight preference) may propagate into the pseudo-labels and cap generalization.

2. The paper employs two scoring schemes DrugCLIP similarity and Vina affinity, but provides insufficient analysis of why, in different settings, DC performs better (DEKOIS 2.0) while Vina prevails (LIT-PCBA).

3. The optimal mixing ratio of generated data must be carefully tuned—more is not always better, limiting the overall gain.

4. SBDD model choice is limited to MolCRAFT; no ablation on alternatives (DiffSBDD, Pocket2Mol, etc.) or sensitivity analysis is provided.

5. Core contribution is standard data-augmentation via generation, with incremental novelty.

**Questions:**

Refer to the weakness.

**Details Of Ethics Concerns:**

NA.

---

> ### Author Response · Authors · 2025-11-20
> **Response to Weaknesses 1,2,3**
>
> We sincerely thank you for your thorough review, insightful comments, and positive recognition of our core idea. We understand the your concerns regarding methodological details and contributions, and we believe our responses below will address these points and further strengthen our work.
> >**W1：AutoDock Vina’s known biases (e.g., molecular-weight preference) may propagate into the pseudo-labels and cap generalization.**
>
> **A1:** We appreciate your observation regarding AutoDock Vina's known biases, such as its preference for higher molecular weight molecules, and the potential for these biases to propagate into our pseudo-labels and cap generalization. We respond to this concern with two main points:
>
> 1.Our experimental results in Tables 1-3 demonstrate that $GenDrugCLIP_{Vina}$ outperform baselines. This superior performance indicates that selecting samples according to vina scores are beneficial for our model's learning process. Furthermore, our method incorporates a stratified sampling step which "coarsely" samples positive and negative examples rather than learning the precise numerical values of the Vina scores. This coarse-grained categorization inherently alleviate the impact of biases from the Vina scoring function.
>
> 2.All existing computational affinity scoring functions, including widely used ones like AutoDock Vina, possess inherent biases and cannot perfectly and unbiasedly reflect true affinities. In practical drug discovery, scientists emphasize relative high scores, plausible binding modes, and comprehensive pharmaceutical properties. Similarly, we filter generated molecules based on drug-likeness and other relevant properties. Crucially, our stratified sampling strategy aims to coarsely distinguish "potentially binding" from "potentially non-binding" categories. This design choice aims to reduce the model's propensity to overfit to the biases of specific scoring function, thereby enhancing robustness and generalization capabilities when meeting novel, unseen molecules.
>
> >**W2:The paper employs two scoring schemes DrugCLIP similarity and Vina affinity, but provides insufficient analysis of why, in different settings, DC performs better (DEKOIS 2.0) while Vina prevails (LIT-PCBA).**
>
> **A2:** We agree with you that a deeper analysis of the differential performance of models across benchmarks would be beneficial. We observed that DrugCLIP exhibits varying performance across different datasets, specifically in terms of BEDROC: DEKOIS 2.0 > DUDE > LIT-PCBA. This indicates that DrugCLIP's intrinsic ability to discriminate between active and inactive molecules, and thus the reliability of its similarity scores, is highly context-dependent. In contrast, AutoDock Vina, as a general-purpose affinity estimation function, tends to offer a more robust and universally applicable measure of binding affinity across a wider range of targets. Our results suggest that when baseline DrugCLIP model demonstrates high performance on a dataset, leveraging its internal similarity metric for pseudo-label generation can further enhance performance. However, on datasets where DrugCLIP’s performance is relatively low, using the more general and robust affinity estimator, Vina, is preferable. This understanding is crucial for selecting the appropriate affinity proxy function to optimize performance in specific drug discovery contexts.
>
> >**W3: The optimal mixing ratio of generated data must be carefully tuned—more is not always better, limiting the overall gain.**
>
> **A3:** We agree with you that "the optimal mixing ratio of generated data must be carefully tuned". This observation is indeed central to effectively leveraging synthetic data without introducing detrimental noise or biases. Even with advanced generative models like MolCRAFT, which achieves reference-level Vina scores for molecules of comparable sizes, not all generated data is equally beneficial for model learning. Introducing noisy or misleading generated molecules can be harmful, hindering generalization. To mitigate this risk, we employ two strategies: stringent filtering and careful control of their mixing ratio with real data. **To empirically determine the optimal balance, we conducted a ablation study on different mixing ratio (e.g., 4:0,4:1,4:2,4:4), as detailed in Section 3.3 of our paper.** This study systematically explored various ratios of generated to real data. The results demonstrated that a 4:2 ratio (gen:real samples) yielded the best overall performance across our benchmarks. Generated positives are most effective when used to enrich rather than dominate the training process. We believe there is potential to further enhance the utility of generated data. This could involve increasing the number of generated molecules or by exploring advanced training strategies such as assigning different learning parameters or weights to supplementary generated molecules, allowing the model to more effectively integrate a larger diverse synthetic data.

---

> ### Author Response · Authors · 2025-11-20
> **Response to Weaknesses 4&5**
>
> >**W4: SBDD model choice is limited to MolCRAFT; no ablation on alternatives (DiffSBDD, Pocket2Mol, etc.) or sensitivity analysis is provided.**
>
> **A4:** Thank you for this valuable point. MolCRAFT offers significant advantages, being **over 30 times faster** than traditional diffusion-based models and **the first to achieve reference-level Vina Score** generation quality with comparable molecular size. These characteristics make MolCRAFT suitable for large-scale, high-quality generation. GenDrugCLIP is designed as a general and modular framework. MolCRAFT serves as an **proof-of-concept** for our core idea.
>
> To further validate the generality of our pipeline, **we conducted additional ablation studies by replacing the SBDD model with Targetdiff.** We set up three distinct experiments: 1. Targetdiff (pos-only): generate only positives using  Targetdiff. 2. Targetdiff (pos&neg): consistent with our main model's setup, but with different model. 3. MC (pos) & TD (neg): molecules generated by MolCRAFT are used as positives, while those generated by Targetdiff serve as negatives. The results are below:
>
> | Ablation | Method | AUROC(%) | EDROC(%) | EF0.5% | EF1% | EF 5% |
> | :---: | :--- | :---: | :---: | :---: | :---: | :---: |
> | – | DrugCLIP | **78.68** | 37.76 | 28.18 | 23.48 | 8.73 |
> | – | $GenDrugCLIP_{Vina}$ | 78.11 | **45.42** | 35.63 | **29.24** | **9.79** |
> | SBDD | Targetdiff(pos-only) | 77.59 | 38.74 | 30.55 | 24.36 | 8.69 |
> |  | Targetdiff(pos&neg) | 77.88 | 42.73 | 33.69 | 27.39 | 9.30 |
> |  | MC(pos) & TD(neg) | 77.73 | 45.27 | **36.23** | 28.93 | 9.73 |
>
> Targetdiff(pos&neg) shows that GenDrugCLIP pipeline still yield comparable improvements over baseline when utilizing the SBDD model TargetDiff. This result shows the generalizability of our pipeline. Since the quality of molecules generated by TargetDiff is weaker than that of MolCRAFT, the model's performance gain from positive samples is lower (Targetdiff(pos-only)). The experiment MC(pos) & TD(neg) achieves results comparable to our main model. This suggests that sampling negative examples from weaker generators is also a viable strategy. By comparing the results of Targetdiff(pos&neg) and $GenDrugCLIP_{Vina}$, sample pseudo positives from stronger model improve overall performance, which implies more performance gain with stronger SBDD model in the future.
>
> >**W5:Core contribution is standard data-augmentation via generation, with incremental novelty.**
>
> **A5:** We appreciate your attention to novelty. We wish to clarify that our work embodies a paradigm shift and holds implications far beyond traditional data augmentation. It directly addresses critical needs within the domain of drug discovery and possesses immense potential. We would like to show the significance of our method through following two points:
>
> 1.**"Learning from synthetic data"** has emerged as a rapidly developing and influential research theme. Numerous studies enhance models in data-scarce domains through synthetic data generation in many fields such as robotics[1] and medical imaging[2]. GenDrugCLIP represents a concrete and innovative instantiation of this trend within drug discovery. By integrating structural biology information into the generation process, it provides high-quality structured knowledge for protein-ligand representation learning. This fundamentally distinguishes our approach from generic data augmentation; it constitutes a novel paradigm of domain-knowledge-driven synthetic data for representation learning.
>
> 2.**GenDrugCLIP is a complete, closed-loop representation learning framework.** The modular design allows for flexible substitution  of its components with more advanced models, adapting to diverse practical scenarios. GenDrugCLIP is orthogonal to existing CLIP-style virtual screening models (e.g., LigUnity[3]) and optimized methods (e.g., DrugHush[4]), suggesting significant potential for synergistic integration. More profoundly, GenDrugCLIP's potential lies in offering solutions for few-shot and zero-shot learning challenges in drug screening. By leveraging the principle of "learning from synthetic data," our approach can generate "virtual" ligands highly matched to novel targets. This effectively augments the available training data, thereby enhancing the model's learning capabilities in these challenging scenarios. Such an advancement holds immense practical significance for accelerating new drug development.
>
> **References**
>
> [1] Chang, Mincheol, et al. "Just Add $100 More: Augmenting Pseudo-LiDAR Point Cloud for Resolving Class-imbalance Problem." NeurIPS 2024.
>
> [2] Li, Jolina, et al. "Infusing synthetic data with real-world patterns for zero-shot material state segmentation." NeurIPS 2024.
>
> [3] Feng, Bin, et al. "Hierarchical affinity landscape navigation through learning a shared pocket-ligand space." Patterns 6.10 (2025).
>
> [4] Han, Jin, Yun Hong, and Wu-Jun Li. "DrugHash: Hashing Based Contrastive Learning for Virtual Screening." AAAI 2025.

---

> > ### Comment · Reviewer_we3Y · 2025-11-26
> >
> > Thanks for the response, the W1 W2 W4 have been basically solved.
> >
> > For W3：the performance gain from the method seems limited to me, so I feel W5 is somewhat overclaimed. Overall, I am open to raising the score, but I will take the other reviewers’ comments into account and reassess more objectively.

---

> ### Author Response · Authors · 2025-11-26
>
> Thank you very much for your prompt and positive feedback.
>
> We hope that, by using the idea that we propose in this work, further performance gain can be achieved when more powerful generative models emerge.

---

### Official Review · Reviewer_XyZo · 2025-10-30

**Soundness:** 3
**Presentation:** 3
**Contribution:** 2
**Rating:** 4
**Confidence:** 5

**Summary:**

This paper proposes a data augmentation method for protein pocket-small molecule contrastive learning called GenDrugCLIP, using the generative ability of SBDD models like MolCRAFT. The method addresses challenges such as the scarcity of true binding data and the use of trivial negative samples by generating target-aware pseudo positives and hard negatives. Experimental results show that GenDrugCLIP outperforms DrugCLIP on multiple benchmarks.

**Strengths:**

1. The idea of using generated model to generated synthetic data for VS task is a great idea, as the real data is hard to get because of the expense of wet lab exps.
2. This paper mentioned a important problem is that the negative sample need to be hard to push the model learn the really important pricinples.

**Weaknesses:**

The main contribution of this paper is focused on the Data Augmentation method, which may not be a very broad topic, and whether it is sufficient for publication at ICLR might be worth discussing. However, I think it’s still quite ok, as data augmentation methods are currently much needed in this field.

For others, see the questions.

**Questions:**

1.	In your motivation, you mentioned that a key challenge is the limited number of true active ligands for each target. Why is this considered a critical issue when modeling pocket-ligand interactions?
2.Have you experimented with combining both the DC score and Vina score filters? If applied simultaneously, do you think this would improve the results?

3.	The baseline results on DUD-E seem to differ from those reported in previous papers, such as DrugCLIP. Is this due to dataset deduplication based on UniProt IDs or another factor?
4.	MolCRAFT is trained using CrossDocked, which is not deduplicated from the test sets. Could using MolCRAFT to generate training data result in data leakage?
5.	It’s not clear how much useful information is provided by MolCRAFT versus the multiple filters. I recommend adding an ablation study where random molecules are first sampled (with or without matching ligand atom numbers to the reference ligand), then subjected to Vina docking to get their poses. Afterward, apply the same property and DC or Vina score filters to select the synthetic data. Would this approach potentially enhance DrugCLIP’s performance?
6.	The distribution of the generated molecules differs from that of real molecules. Could this pose a problem for your approach?

---

> ### Author Response · Authors · 2025-11-20
> **Response to Weakness 1 & Question 1(1)**
>
> We sincerely thank you for your thorough review, insightful comments, and positive recognition of our core idea as “a great idea”. We provide our responses below.
>
> >**W1:The main contribution of this paper is focused on the Data Augmentation method, which may not be a very broad topic, and whether it is sufficient for publication at ICLR might be worth discussing. However, I think it’s still quite ok, as data augmentation methods are currently much needed in this field.**
>
> **A1:** We sincerely appreciate your insightful observation that data augmentation methods are currently much needed in this field. Regarding the broad impact of our method, we wish to clarify that our work embodies a paradigm shift and holds implications far beyond traditional data augmentation. We would like to elaborate on the significance of our method through the following two points:
>
> 1.**"Learning from synthetic data"** has emerged as a rapidly developing and highly influential research theme. Numerous studies are dedicated to enhancing model in data-scarce domains through synthetic data generation and augmentation strategies in many fields such as robotics[1] and medical imaging[2]. GenDrugCLIP represents a concrete and innovative instantiation of this trend within drug discovery. By deeply integrating structural biology information into the generation process, it provides high-quality structured knowledge for protein-ligand representation learning. This fundamentally distinguishes our approach from generic data augmentation; it constitutes a novel paradigm of domain-knowledge-driven synthetic data for representation learning.
>
> 2.**GenDrugCLIP establishes a complete, closed-loop representation learning framework.** The Generate-Filter-Score-Select modular design allows for flexible substitution or optimization of its components with more advanced models, adapting to diverse practical scenarios. GenDrugCLIP's methodology is orthogonal to existing CLIP-style virtual screening models (e.g., LigUnity[3]) and optimization methods (e.g., DrugHush[4]), suggesting significant potential for synergistic integration. More profoundly, GenDrugCLIP's potential lies in offering solutions for few-shot and zero-shot learning challenges in drug screening. By leveraging the principle of "learning from synthetic data," our approach can generate "virtual" ligands highly matched to novel targets. This effectively augments the available training data, thereby enhancing the model's learning capabilities in these challenging scenarios. Such an advancement holds immense practical significance for accelerating new drug development.
>
> **References**
>
> [1] Chang, Mincheol, et al. "Just Add $100 More: Augmenting Pseudo-LiDAR Point Cloud for Resolving Class-imbalance Problem." Advances in Neural Information Processing Systems 37 (2024): 66226-66259.
>
> [2] Li, Jolina, Manuel Drehwald, and Alan Aspuru-Guzik. "Infusing synthetic data with real-world patterns for zero-shot material state segmentation." Advances in Neural Information Processing Systems 37 (2024): 60237-60259.
>
> [3] Feng, Bin, et al. "Hierarchical affinity landscape navigation through learning a shared pocket-ligand space." Patterns 6.10 (2025).
>
> [4] Han, Jin, Yun Hong, and Wu-Jun Li. "DrugHash: Hashing Based Contrastive Learning for Virtual Screening." Proceedings of the AAAI Conference on Artificial Intelligence.
> >**Q1(1): In your motivation, you mentioned that a key challenge is the limited number of true active ligands for each target. Why is this considered a critical issue when modeling pocket-ligand interactions?**
>
> **A2:** Thank you for this insightful question, which highlights a fundamental challenge in the field.
> When a target has only a limited number of known active ligands, the model is exposed to a very narrow, and often biased, portion of the "binding chemical space" relevant to that target. This severely hinders the model's ability to learn generalizable interaction patterns. It makes the model susceptible to overfitting to specific ligand chemical scaffolds and binding modes that happen to be present within the limited training data. In PDBbind, a single target corresponds to an average of only 3-4 known ligands. This extreme sparsity of diverse active ligands is precisely the challenge our work aims to address. Our method strategically generates target-aware, high-quality pseudo-positive ligand molecules. This process effectively expands the observed chemical diversity and provides the model with a richer and more varied set of examples, thereby alleviating the issues arising from the scarcity of real active ligands and enabling it to learn more robust and generalizable pocket-ligand interaction rules.

---

> ### Author Response · Authors · 2025-11-20
> **Response to Questions 1(2), 2,3**
>
> >**Q1(2): Have you experimented with combining both the DC score and Vina score filters? If applied simultaneously, do you think this would improve the results?**
>
> **A3:** Thank you for this excellent suggestion, which prompted further valuable investigation into our sample selection strategy. We conducted two new experiments on DUD-E benchmark for positive sample selection, while keeping the negative sample selection unchanged. These new criteria were:
> Setting 1 (low $S_V$ & high $S_D$): Samples deemed 'good' by both Vina and DrugCLIP, and Setting 2 (low $S_V$ & low $S_D$): Samples where Vina indicated good physical binding but DrugCLIP indicated low similarity are regarded as positive samples.
>
> | Method | AUROC(%) | EDROC(%) | EF0.5% | EF1% | EF 5% |
> | :--- | :---: | :---: | :---: | :---: | :---: |
> | $GenDrugCLIP_{Vina}$ | 78.11 | 45.42 | 35.63 | 29.24 | 9.79 |
> | Setting1 | 77.49 | 44.77 | 35.01 | 29.06 | 9.62 |
> | Setting2 | **79.73** | **46.31** | **36.00** | **29.86** | **10.06** |
>
> Our addtional experiments show that Setting 2 modestly improved our model's performance. The rationale behind this likely lies in the fact that these samples, despite being considered 'poor' by DrugCLIP (low $S_D$), possess good physical binding characteristics according to Vina (low $S_V$). Such samples represent valuable 'challenging positives' that, when incorporated, compel the model to refine its understanding of relevant interactions and learn more nuanced and robust representations. Conversely, Setting 1 (low $S_V$ & high $S_D$), yields inferior performance over $GenDrugCLIP_{Vina}$. This suggests that these 'easy' positive examples, which are already well-aligned between physical binding (Vina score) and semantic similarity (DrugCLIP score), provides less novel information.
> Combining two scoring metrics offers comprehensive assessment, mitigating biases inherent in relying on a single scoring method. In addition, the strict criteria of combining both scores reduced the number of available samples for training. While the overall concept of leveraging multiple perspectives for sample selection is sound, the limited sample size constrained the observed improvement in these experiments, suggesting that the benefits might be more pronounced when a larger pool of such combined-criterion samples are generated.
>
> >**Q2: The baseline results on DUD-E seem to differ from those reported in previous papers, such as DrugCLIP. Is this due to dataset deduplication based on UniProt IDs or another factor?**
>
> **A4:** You are correct. The diffrence in baseline results stem from our adoption of the same deduplication strategy as employed in EquiScore. The deduplication of PDB id in DrugCLIP is too weak and protein-ligand pair of the same protein are leaked to training set. We have discussed the impact of duplicate targets and our deduplication method in detail in Section 3.1 and further elaborated in Appendix C. Conceretely, we re-trained the model on deduplicated dataset, and this re-trained version is referred to as $DrugCLIP_{dedup}$ ("DrugCLIP" in the main part of our paper). For all other baselines reported in our work, their results are consistent with those presented in EquiScore publication. This consistent application of a deduplication process ensures a fair and comparable evaluation across all models within our study.
>
> >**Q3: MolCRAFT is trained using CrossDocked, which is not deduplicated from the test sets. Could using MolCRAFT to generate training data result in data leakage?**
>
> **A5:** Thank you for raising this critical and insightful concern regarding potential data leakage.We address the risk of data leakage with the following three points:
>
> **1. No Direct Access to Test Set Pockets:** Our methodology ensures that MolCRAFT is used to generate molecules only for pockets in training set. Crucially, under the stringent deduplication settings adopted in our work, MolCRAFT has no access to information that overlap with the test sets based on UniProt IDs. This prevents any direct data leakage from test set pockets into the generation process for our training data.
>
> **2. Theoretical Indirect Leakage is Minimised:** While there exists a theoretical possibility that MolCRAFT, trained on the original CrossDocked dataset, might have learned patterns from pocket-ligand pairs in the test set, and could indirectly manifest these patterns when generating molecules for training set pockets. We argue this is highly improbable to cause leakage. Ideally, MolCRAFT itself would be trained on a CrossDocked dataset that is deduplicated against the test sets. However, the proportion of proteins in CrossDocked that share UniProt IDs with our test sets is less than 5%. Given this small overlap and the fact that MolCRAFT's generation is conditioned on specific training set pockets, it is highly unlikely to favor or "recall" features specifically tied to these few overlapping test set proteins.
>
> **Our response to Q3 continues in the next box.**

---

> ### Author Response · Authors · 2025-11-20
> **Response to Questions 3,4,5**
>
> **3. Empirical Evidence Rules Out Leakage:** To definitively address the concern of data leakage and  assess the actual impact of any potential "information leakage," we calculate the Tanimoto similarity between the generated molecules (which are used to train our model) and the ligands present in test sets. In a pairwise similarity evaluation, fewer than 0.001% of the generated molecules exhibit a Tanimoto similarity greater than 0.5 with any molecule in the test sets. This extremely low overlap indicates that the generated molecules are structurally distinct from the actual test set ligands, effectively ruling out any significant structural data leakage into our training data.
>
> >**Q4: It’s not clear how much useful information is provided by MolCRAFT versus the multiple filters. I recommend adding an ablation study where random molecules are first sampled (with or without matching ligand atom numbers to the reference ligand), then subjected to Vina docking to get their poses. Afterward, apply the same property and DC or Vina score filters to select the synthetic data. Would this approach potentially enhance DrugCLIP’s performance?**
>
> **A6:** We deeply appreciate this insightful suggestion for an ablation study. It directly addresses the core value proposition of MolCRAFT's target-aware generation versus a more generic 'generate and filter' approach.
>
> We previously conducted a related experiment in Table 4 as the "with trivial Neg". We replaced our carefully selected hard negative samples with "random molecules from ZINC." This led to a significant drop in model performance (BEDROC: 39.02%) compared to main model (45.42%). However, that experiment did not fully isolate the contribution from the subsequent multi-stage filtering process.
>
> Your suggested experiment offers a more rigorous and direct control to evaluate the contribution of MolCRAFT's generation. Therefore, we conduct an additional ablation study on DUD-E, which we refer to as Setting 3.
> In Setting 3, we randomly sample molecules from ZINC (match the ligand atom numbers to reference ligands) to ensure a fair comparison in terms of molecular size. These randomly sampled molecules are subjected to the same property filters and DrugCLIP scoring as applied in our main framework. The distinction between Setting 3 and $GenDrugCLIP_{DC}$ lies solely in the source of data. This allows us to assess the value added by MolCRAFT's generative capabilities. The results for this new Setting 3 are presented below:
>
> | Method | AUROC(%) | EDROC(%) | EF0.5% | EF1% | EF 5% |
> | :--- | :---: | :---: | :---: | :---: | :---: |
> | DrugCLIP | 78.68 | 37.76 | 28.18 | 23.48 | 8.73 |
> | $GenDrugCLIP_{DC}$ | 77.80 | 43.60 | 34.99 | 28.07 | 9.38 |
> | Setting3 | 75.69 | 38.68 | 30.67 | 24.39 | 8.57 |
>
> The results indicate that the model based on randomly sampled molecule has almost no improvement compared to baseline, and randomly molecules are difficult to provide effective learning information to the model without pocket information.
>
> >**Q5:The distribution of the generated molecules differs from that of real molecules. Could this pose a problem for your approach?**
>
> **A7:** Thank you for raising this pertinent question regarding the distributional differences between generated and real molecules.
> We recognize that an over-reliance on generated data could potentially cause the model to learn biases inherent in the generative model. To mitigate this, we control the proportion of generated molecules mixed with real molecules during training. In our paper, Section 3.3 details an ablation study specifically designed to determine the optimal mixing ratio. We experimented with various ratios (e.g., 4:0, 4:1, 4:2, 4:4 for gen vs. real samples) and found that a 4:2 ratio yielded the best performance. This controlled mixing effectively modulates how much the model relies on the chemical space provided by generated molecules, striking a balance between leveraging novelty and preventing the dominance of generative biases.
>
> Crucially, the distributional difference between generated molecules and the real molecules in the training data is a desired and actively utilized characteristic of our framework. We leverage the novelty introduced by the generative model to expand the chemical space. Specifically, the pseudo-positive samples are designed to fill in the sparsity of the chemical space around known active ligands, providing the model with a more comprehensive set of learning examples within biologically relevant regions. The hard negative samples are crafted to be structurally similar to actives but are inactive, forcing the model to learn finer, more discriminative features that differentiate true binders from decoys. All generated molecules undergo drug-likeness and property filtering. This step ensures that we only introduce molecules with favorable drug-like characteristics, reducing the risk of incorporating irrelevant or harmful distributions into the training process.

---

### Official Review · Reviewer_oY5n · 2025-11-04

**Soundness:** 3
**Presentation:** 3
**Contribution:** 2
**Rating:** 4
**Confidence:** 5

**Summary:**

The paper introduces GenDrugCLIP, a generation-augmented framework for contrastive drug–target representation learning. The key idea is to repurpose structure-based drug design models as controllable data engines that can generate target-aware molecules. GenDrugCLIP employs a Generate–Filter–Score–Select pipeline to produce pseudo positives (high-scoring molecules) and hard negatives (low-scoring molecules) for triplet-based contrastive learning. Experiments on three benchmarks show that GenDrugCLIP outperforms existing methods such as DrugCLIP.

**Strengths:**

- Creative reuse of SBDD models as generative data engines, bridging generative modeling and contrastive learning.
- Demonstrated strong empirical gains over DrugCLIP across multiple benchmarks.
- Conceptually broadens the role of generative models in drug discovery from candidate generation to data augmentation for representation learning.

**Weaknesses:**

- Compared to DrugCLIP, the methodological novelty is limited. The main contribution appears to be the generation of an augmented dataset rather than a fundamentally new learning framework.
- Only one SBDD method is used for data augmentation, and it is unclear why this specific method was chosen or how robust the approach would be if alternative SBDD generators were applied.
- ground truth positive compounds might not even pass a strict filter like the one the authors implemented here, potentially introducing a data distribution shift between the augmented and original datasets.
- The paper does not discuss how sensitive or robust the model is to imperfect or less curated filtering procedures, which could affect generalization in practical scenarios.

**Questions:**

- GenDrugCLIP consistently outperforms both DrugCLIP and EquiScore across most metrics, but the differences in AUROC are relatively small. Does this indicate that GenDrugCLIP primarily improves early enrichment rather than overall ranking performance?

---

> ### Author Response · Authors · 2025-11-20
> **Response to Weaknesses 1&3**
>
> We sincerely thank you for your thorough reading, insightful comments, and constructive feedback on our manuscript. We are very grateful for your recognition of our work's methodological creativity, strong empirical gains and conceptual broadening. We provide our responses below.
>
> >**W1: Compared to DrugCLIP, the methodological novelty is limited. The main contribution appears to be the generation of an augmented dataset rather than a fundamentally new learning framework.**
>
> **A1:** We appreciate your attention to novelty. We wish to clarify that our work embodies a paradigm shift and holds implications far beyond traditional data augmentation. It directly addresses critical needs within the domain of drug discovery and possesses immense potential. We would like to show the significance of our method through the following two points:
>
> 1.**"Learning from synthetic data"** has emerged as a rapidly developing and highly influential research theme. Numerous studies are dedicated to enhancing model in data-scarce domains through synthetic data generation and augmentation strategies in many fields such as robotics[1] and medical imaging[2]. GenDrugCLIP represents a concrete and innovative instantiation of this trend within drug discovery. By deeply integrating structural biology information into the generation process, it provides high-quality structured knowledge for protein-ligand representation learning. This fundamentally distinguishes our approach from generic data augmentation; it constitutes a novel paradigm of domain-knowledge-driven synthetic data for representation learning.
>
> 2.**GenDrugCLIP establishes a complete, closed-loop representation learning framework.** The Generate-Filter-Score-Select modular design allows for flexible substitution or optimization of its components with more advanced models, adapting to diverse practical scenarios. GenDrugCLIP's methodology is orthogonal to existing CLIP-style virtual screening models (e.g., LigUnity[3]) and optimization methods (e.g., DrugHush[4]), suggesting significant potential for synergistic integration. More profoundly, GenDrugCLIP's potential lies in offering solutions for few-shot and zero-shot learning challenges in drug screening. By leveraging the principle of "learning from synthetic data," our approach can generate "virtual" ligands highly matched to novel targets. This effectively augments the available training data, thereby enhancing the model's learning capabilities in these challenging scenarios. Such an advancement holds immense practical significance for accelerating new drug development.
>
> **References**
>
> [1] Chang, Mincheol, et al. "Just Add $100 More: Augmenting Pseudo-LiDAR Point Cloud for Resolving Class-imbalance Problem." Advances in Neural Information Processing Systems 37 (2024): 66226-66259.
>
> [2] Li, Jolina, Manuel Drehwald, and Alan Aspuru-Guzik. "Infusing synthetic data with real-world patterns for zero-shot material state segmentation." Advances in Neural Information Processing Systems 37 (2024): 60237-60259.
>
> [3] Feng, Bin, et al. "Hierarchical affinity landscape navigation through learning a shared pocket-ligand space." Patterns 6.10 (2025).
>
> [4] Han, Jin, Yun Hong, and Wu-Jun Li. "DrugHash: Hashing Based Contrastive Learning for Virtual Screening." Proceedings of the AAAI Conference on Artificial Intelligence.
>
> >**W3: ground truth positive compounds might not even pass a strict filter like the one the authors implemented here, potentially introducing a data distribution shift between the augmented and original datasets.**
>
> **A3:** We appreciate your comment regarding potential data distribution shifts due to filtering.
> It is true that a portion of ground truth (GT) positive compounds might not pass our drug-likeness filter. However, the primary objective of this filter is not to strictly maintain the exact distribution of the original dataset, but rather to prevent the introduction of detrimental data caused by "unrealistic" or "undesirable" molecules often generated by SBDD models. Therefore, data distribution shifts due to filtering is acceptable in this work.
>
> We employ widely recognized metrics for filtering, including QED, SA, LogP, and Lipinski’s Rule of Five. While these metrics do not serve as direct optimization targets for virtual screening tasks themselves, they effectively inject knowledge about crucial downstream pharmaceutical properties into the early, large-scale screening process. This proactive integration of drug-likeness criteria is designed to positively impact the protracted and resource-intensive drug discovery pipeline.

---

> ### Author Response · Authors · 2025-11-20
> **Response to Weaknesses 2&4**
>
> >**W2: Only one SBDD method is used for data augmentation, and it is unclear why this specific method.**
>
> **A2:** Thank you for this valuable point. MolCRAFT offers significant advantages, being **over 30 times faster** than traditional diffusion-based models and **the first to achieve reference-level Vina Score** generation quality with comparable molecular size. These characteristics make MolCRAFT suitable for large-scale, high-quality generation. Furthermore, GenDrugCLIP is designed as a general and modular framework. MolCRAFT serves as an excellent **proof-of-concept** for demonstrating the power of repositioning SBDD models as data engines.
>
> To further address your concern and validate the generality of our pipeline, **we conducted additional ablation studies on DUD-E by replacing the SBDD model with Targetdiff.** We set up three distinct experiments: 1. Targetdiff (pos-only): generate only positive samples using  Targetdiff. 2. Targetdiff (pos&neg): consistent with our main model's setup, but with a different SBDD model. 3. MC (pos) & TD (neg): molecules generated by MolCRAFT are used as positive samples, while those generated by Targetdiff serve as negative samples. $GenDrugCLIP_{Vina}$ refers to our main model. The experimental results are presented in the table below:
>
> | Ablation | Method | AUROC(%) | EDROC(%) | EF0.5% | EF1% | EF 5% |
> | :---: | :--- | :---: | :---: | :---: | :---: | :---: |
> | – | DrugCLIP | **78.68** | 37.76 | 28.18 | 23.48 | 8.73 |
> | – | $GenDrugCLIP_{Vina}$ | 78.11 | **45.42** | 35.63 | **29.24** | **9.79** |
> | SBDD | Targetdiff(pos-only) | 77.59 | 38.74 | 30.55 | 24.36 | 8.69 |
> |  | Targetdiff(pos&neg) | 77.88 | 42.73 | 33.69 | 27.39 | 9.30 |
> |  | MC(pos) & TD(neg) | 77.73 | 45.27 | **36.23** | 28.93 | 9.73 |
>
> By comparing the results of Targetdiff(pos&neg) and DrugCLIP, we can see that GenDrugCLIP pipeline still yield comparable improvements over baseline when utilizing the SBDD model TargetDiff. This result demonstrates the generalizability of our pipeline. Since the quality of molecules generated by TargetDiff is weaker than that of MolCRAFT, the model's performance gain from positive samples is lower (Targetdiff(pos-only)). Intriguingly, the experiment MC(pos) & TD(neg) achieves results comparable to our main findings. This suggests that sampling negative examples from weaker generators is also a viable strategy. By comparing the results of Targetdiff(pos&neg) and $GenDrugCLIP_{Vina}$, we can see that sample pseudo positives from stronger model improve performance, which implies more performance gain with stronger SBDD model in the future.
>
> >**W4: The paper does not discuss how sensitive or robust the model is to imperfect or less curated filtering procedures, which could affect generalization in practical scenarios.**
>
> **A4:** Thank you for this insightful question on the impact and robustness of our filtering strategy.
>
> Our selection pipeline involves three main stages: Filter-Score-Select. Regarding the 'Score' stage, we have already presented studies for two distinct scoring methods in Tables 1-3 in the main paper, demonstrating that both functions contribute positively to pipeline. To investigate the impact of the 'Filter' stage, we conducted additional ablation experiments on DUD-E benchmark. These included scenarios where the filtering step was removed ('No Filtering') and where a loose Vina score filtering was applied (allowing 90% of GT molecules to pass, termed 'Loose Filtering').
>
> For the 'Select' stage, our GenDrugCLIP_Vina model typically employs a sampling strategy where positive samples are defined by a Vina score less than -10 ($S_V$ < -10) and negative samples by $S_V$ > -8. We have now performed comparative experiments to assess the influence of different sampling thresholds on the selection process.
>
> | Ablation | Method | AUROC(%) | EDROC(%) | EF0.5% | EF1% | EF 5% |
> | :--- | :--- | :---: | :---: | :---: | :---: | :---: |
> | – | $GenDrugCLIP_{Vina}$ | 78.11 | **45.42** | **35.63** | **29.24** | 9.79 |
> | Filter | No Filtering | 78.35 | 43.88 | 33.98 | 27.93 | 9.60 |
> | | Loose Filtering | 78.53 | 43.82 | 34.39 | 28.26 | 9.53 |
> | Sample | pos<-12 neg>-8 | **79.92** | 44.06 | 34.02 | 27.58 | **9.91** |
> | | pos<-8 neg>-8 | 79.02 | 43.68 | 33.69 | 27.74 | 9.62 |
> | | pos<-10 neg>-10 | 77.75 | 42.49 | 33.51 | 27.31 | 9.27 |
> | | pos<-10 neg>-5 | 77.92 | 44.78 | 35.61 | 28.96 | 9.66 |
>
> Our findings indicate that both 'No Filtering' and 'Loose Filtering' exhibited slightly weaker performance compared to the full pipeline in term of BEDROC and EF, which are important metrics for virtual screening. Nevertheless, they still significantly outperformed the baseline DrugCLIP model. This demonstrates that even with imperfect or less curated filtering criteria, the performance advantage of our method remains substantial, highlighting the robustness of our overall pipeline with respect to the specific filtering choices.
>
> **Our response to W4 continues in the next box.**

---

> > ### Author Response · Authors · 2025-11-20
> > **Response to Weakness 4 and Question 1**
> >
> > In the ablation study for sampling thresholds: Setting a more stringent positive threshold ('pos<-12, neg>-8') for selecting higher-affinity samples did not yield performance improvements over our main method. This is likely because overly strict sample selection can introduce an excessive bias towards high Vina scores and drastically reduce the number of eligible molecules for training.
> > Conversely, removing the 'gap zone' between positive and negative samples ('pos<-8, neg>-8' or 'pos<-10, neg>-10') lead to a decrease in performance. This suggests that the introduction of a 'gap zone' is crucial, as it effectively minimizes the impact of ambiguous samples (those with Vina scores near the decision boundary) on model training. A slightly wider separation between positive and negative thresholds ('pos<-10, neg>-5') achieved performance closest to our main model. This indicates that for target-aware negative sampling, it is not always necessary to select only the 'hardest' or most similar negative samples. Slightly weaker, yet still target-aware, negative samples can also provide valuable signals for learning.
> >
> > >**Q1：GenDrugCLIP consistently outperforms both DrugCLIP and EquiScore across most metrics, but the differences in AUROC are relatively small. Does this indicate that GenDrugCLIP primarily improves early enrichment rather than overall ranking performance?**
> >
> >
> > **A5:** Thank you for this insightful question, which highlights a key aspect of our model's performance. Indeed, GenDrugCLIP primarily focuses on and significantly improves early enrichment. The ultimate goal of virtual screening is to efficiently and accurately identify a small number of highly potent active molecules from a vast chemical space. Metrics such as BEDROC and Enrichment Factor (EF) are precisely the most direct and crucial measures of this capability.
> >
> > In this work, our model demonstrates a notable improvement in $EF_{0.5\\%}$ on the DUD-E benchmark from 28.18% to 35.63% (+7.45%). In an ideal drug screening scenario, this translates to significant practical benefits. With the same wet-lab experimental investment, our model is expected to discover approximately 26% more active compounds. Alternatively, to obtain the same number of active compounds, only 80% of the original wet-lab testing effort would be required, thus conserving valuable experimental resources. An improvement in early enrichment metrics directly translates to discovering more active compounds earlier in the drug discovery pipeline, thereby leading to substantial savings in experimental costs and time.
> >
> > GenDrugCLIP specifically addresses the train-test domain gap identified in DrugCLIP by constructing targeted positive and negative target-aware sample pairs. The relative stability of AUROC suggests that the baseline model already performs well in distinguishing a large proportion of negative samples from positives. The primary contribution of GenDrugCLIP lies in significantly enhancing the model's performance on the most critical subset of candidate molecules that require fine-grained discrimination.

---

> ### Comment · Reviewer_oY5n · 2025-11-26
>
> Thank you for the detailed response. I have raised my scores accordingly

---

> > ### Author Response · Authors · 2025-11-27
> >
> > Thank you for your prompt and positive feedback.
> >
> > We greatly appreciate your careful consideration of our rebuttal and the time you took to re-evaluate our work.

---

### Meta-Review · Area_Chair_6mRU · 2026-01-01

**Summary:**

The paper addresses a practical problem in drug discovery with a well-executed application of SBDD models for data augmentation. Reviewers all appreciated the creative reuse of generative models and solid empirical gains. The rebuttal successfully addressed technical concerns through comprehensive ablations.

However, all reviewers unanimously raised concerns about limited methodological novelty and this fundamental concerns remains largely unresolved after the rebuttal. Based on this, I recommend rejection.

**Reviewer Concerns:**

**Concerns addressed**

1. Generalizability of SBDD Models (oY5n, we3Y, 8bmZ)
> The authors added experiments using TargetDiff as an alternative generator. While performance gains were lower than with MoICRAFT, the pipeline still improved on the baseline

2. Robustness to filtering and sampling strategies (oY5n, XyZo, 8bmZ)
> The concern is addressed by experiments (Setting 1/2/3) and ablations on no filtering, loose filtering, and threshold variation.

3. Data leakage (XyZo)
> The authors conducted a Tanimoto similarity check showing that fewer than 0.001% of generated training molecules had a similarity $> 0.5$ to test set ligands13. They also clarified that MoICRAFT only generated data for training-set pockets.

**Outstanding concerns**
1. Limited methodological novelty; a data augmentation method (all reviewers)
> The authors argue for a “paradigm shift” toward "learning from synthetic data", but this somewhat overclaimed. Two papers from robotics and medical imaging are cited to justify that this makes sufficient contribution for top ML conference. However, the cited medical imaging paper appears in the NeurIPS 2024 Datasets and Benchmarks track rather than the main conference and naturally focuses on data generation. Moreover, that work studies material images rather than medical images, suggesting the citation may not have been carefully examined. In addition, the rebuttal emphasizes potential impact in few-shot or zero-shot scenarios rather than clearly demonstrated contributions. Therefore, this concern remains substantive.

2. Sensitivity and tuning burden (we3Y)
> The ablations show that tuning of mixing ratio is required. The authors confirm Reviewer we3Y’s concern.

**Reviewer Scores:**

All reviewers initially rated the paper as 4. After the rebuttal, most reviewers would likely increase their scores, as many of their questions were addressed. Reviewers oY5n and XyZo are likely to increase their scores to 6. Reviewers we3Y and 8bmZ are likely to remain at 4, as their major concerns persist.

---

### Decision · Program_Chairs · 2026-01-26

Reject